# Current Understanding of Water Properties inside Carbon Nanotubes

**DOI:** 10.3390/nano12010174

**Published:** 2022-01-05

**Authors:** Aris Chatzichristos, Jamal Hassan

**Affiliations:** Department of Physics, Khalifa University, Abu Dhabi 127788, United Arab Emirates

**Keywords:** carbon nanotubes, water, nuclear magnetic resonance, molecular dynamics

## Abstract

Confined water inside carbon nanotubes (CNTs) has attracted a lot of attention in recent years, amassing as a result a very large number of dedicated studies, both theoretical and experimental. This exceptional scientific interest can be understood in terms of the exotic properties of nanoconfined water, as well as the vast array of possible applications of CNTs in a wide range of fields stretching from geology to medicine and biology. This review presents an overreaching narrative of the properties of water in CNTs, based mostly on results from systematic nuclear magnetic resonance (NMR) and molecular dynamics (MD) studies, which together allow the untangling and explanation of many seemingly contradictory results present in the literature. Further, we identify still-debatable issues and open problems, as well as avenues for future studies, both theoretical and experimental.

## 1. Introduction

In recent years, the topic of water and other liquids intercalating and diffusing in carbon nanotubes (CNTs) has attracted significant interest. Water confined in the narrow interior of CNTs exhibits properties distinctly different than its bulk form. Indeed, water’s phase diagram, the topology of its hydrogen network and its diffusivity all heavily depend on the geometry of the CNT and the hydrophobic interactions between water molecules and the CNT walls. Water inside carbon nanotubes has been shown to exhibit a much faster diffusivity than that of bulk for certain CNT diameters and a flow rate orders of magnitude larger than theoretical predictions. These characteristics are important both in terms of fundamentally understanding the associated physics which govern related phenomena in a wide variety of systems across several scientific fields, as well as for a cornucopia of novel applications.

First, studying the properties of water in nanoconfinement can act as a model for understanding the physical mechanisms in a series of more complicated systems. In this context, CNTs are used to study water interacting with hydrophobic surfaces, while porous materials such as silica MCM-41 or SBA-15 [1,2] are typically used for modeling hydrophilic interactions. As the diameter of the CNTs can easily be varied, the effects of dimensional restriction on the properties of water can readily be studied, going from effectively 3D bulk water if the tube’s diameter is large (e.g., d > 10 nm), to restricted 1D diffusion, or even anomalous single-file and stratified motion for smaller diameters (d ∼ 1 nm) [3]. This favorable arrangement has been used to extract important information, since CNTs can serve as models for understanding molecular-level hydrodynamics in more complicated systems such as properties of biological transmembrane channels [4]. Indeed, biological pores and membranes are commonly hydrophobic and are known to regulate the flow of water and several solutes in the cell, frequently with a high degree of selectivity. For these reasons, CNTs are used as a simplified model to understand the underlying mechanisms of the various pore properties [5,6,7,8]. The fields of geology, fluid dynamics and chemical catalysis are also treating CNTs as a toy-model for more complicated systems. As such, water flow in CNTs has been used to model the swelling of clay minerals [9], the much more complicated flow of (multiphase) oil-water mixtures through rock formations and water diffusing through cement, rocks and soil (e.g., through nanoporous zeolites) [10], with these results informing geological models of underground reservoirs. Furthermore, this is an excellent platform for studying water-surface interactions at the nanoscale, which can lead to insights for enhancing the effectiveness of several catalyst nanoparticles [11].

Second, the potential applications of CNTs (dry or water-filled) currently seem nearly limitless. CNTs are proposed as a basis of single-molecule detectors [12,13], conductive and high-strength composites [14], gas and humidity sensors [15,16], terahertz devices [17], electrodes in thin-film photovoltaics [18] and components of energy (including hydrogen) storage systems [19,20,21]. Water-filled narrow CNTs are shown to form a “proton wire” via the unified hydrogen-bond network of the confined water molecules, which is suggested to modulate the relevant proton conductance, offering the possibility of forming a switchable nanoscale semiconductor [22]. CNTs have also been utilized as the electron emitter (cathode) for the X-ray tube of medical imaging scanners (CT-scanners) [23] and nanometer-sized semiconductor devices (as field-effect transistors and in integrated circuit applications) [24]. In addition, CNTs have been shown to simultaneously allow the seamless passage of water molecules, while also reject salt and many nano-pollutants that are present in sea and fresh water. Thus, CNT-based water treatment systems can potentially aid in desalination and pollutant-removal projects, owning also to the self-cleaning function and low energy consumption of CNT membranes [25]. Especially for solar desalination applications, in which the energy for the separation of fresh water from seawater is provided directly by the sun in the form of thermal energy, it has been shown that the interactions between seawater and the filter membrane’s microstructures controls the process’ performance [26].

Expanding our understanding of water (and other liquids) diffusing in CNTs is a key stepping stone for the development of several promising medical applications [27,28], especially since the physio-chemical properties of nanoconfined water influence the biological response to it [29]. Furthermore, the diffusion properties of water near the magnetic resonance imaging (MRI) contrast agent control to a large extent the quality of the imaging [30]. Adding on, CNTs have been proposed as potential drug delivery systems against cancer [31] and Alzheimer’s dementia [27,32], especially since they are known to cross the blood–brain barrier [33]. The fact that several cancer drugs are hydrophobic [31], further enhances the importance of studying liquid-CNT systems in this setting. CNTs are also considered attractive options for potential uses in diagnostics [28], nanosyringes [34] and patient-specific artificial implants [35].

The aforementioned numerous applications of CNTs, as well as the details of their interaction with water molecules, stem from their particular geometrical structure. One can think of a CNT as one (or several) sheet(s) of graphene (with its characteristic 2D hexagonal honeycomb lattice) folded into a tube. Generally, CNTs are split in three categories, depending on how many graphene sheets are used to form the CNT wall. These are noted as single-walled (SWCNTs), double-walled (DWCNTs) and multi-walled carbon nanotubes (MWCNTs), accordingly. The structure of the CNTs is discussed in greater detail in Section 2.

The diameter and length of CNT channels significantly affect the properties of water molecules diffusing inside them. This is due to the hydrophobic nature of the CNTs’ walls, which repel water molecules towards the center of the tube. Most CNTs have impurities, defects and a non-zero surface roughness which add additional molecular interactions with water molecules.

Unlike theoretical works on water-CNT systems, in experimental settings real CNTs generally form bundles of individual tubes. Thus, water can be in bulk form outside the bundles, with little-to-no interaction with the CNT walls, or occupy the interstitial space between bundles, weakly interacting with the exterior of its neighboring CNTs, or enter inside individual CNT channels. In each case, water’s dynamical properties can vary greatly, as its environment is strikingly different in terms of dimensional confinement and proximity to the hydrophobic walls. In addition, in recent years it has been made progressively clearer, that water inside CNTs can form multiple components, such as several concentric (ice-like) water rings and/or a central water chain, each with distinct characteristics in terms of hydrogen-network geometry, diffusive properties and freezing temperature. The existence and specific properties of these water components depend on the diameter of the CNT as well as on thermodynamical variables such as temperature and pressure (see Section 5).

Given the plethora of interesting effects and possible applications of water in CNTs, it is probably not surprising that these systems have attracted great research interest and have been studied with theoretical and experimental techniques. These studies have progressively advance our understanding of the electrical and mechanical properties of CNTs, as well as the nature and characteristics of water in the carbon nanotubes. In this context, the arsenal of experimental methods applied to the system at hand includes X-ray Compton scattering [36,37], transmission electron microscopy (TEM) [38,39,40,41,42], infrared [43], dielectric [44] and Raman [45,46] spectroscopies, thermogravimetric analysis [47] differential scanning calorimetry, dielectric relaxation spectroscopy, neutron diffraction and adsorption [48], as well as and nuclear magnetic resonance (NMR) [49,50,51,52,53,54,55,56,57,58,59,60,61,62,63,64]. From a theoretical standpoint, molecular dynamics (MD) is the most common toolset utilized to study the properties of water in CNTs [4,22,30,65,66,67,68,69,70,71,72,73,74,75,76,77,78,79,80,81,82,83,84,85,86,87,88]. Indeed, most early studies of this system were theoretical, following the pioneer work of Hummer et al. in 2001 [65].

In this review, we will survey the recent research on water in CNTs, by separately discussing each of the several research areas that collectively comprise the aforementioned system. Although we will mention relevant results from all techniques, the main focus will be given to systematic NMR and MD studies, as they yield complementary information on the microscopic/local environment and can also identify the different components of the system, thus providing a unified picture of the characteristics of nanoconfined water. Here we identify seven broad areas that together form all main aspects of water in CNTs: First, there is a fundamental question of whether water can even enter the CNT channels at all, given that the latter are made of hydrophobic graphene sheets and their nano sizes are expected to reduce the entropy of the uptake process. In addition, the surface tension to enter the CNT [49,89] is 0.2 N/m, much larger than the surface tension of water ( 0.072 N/m). Nevertheless, several studies—both theoretical and experimental—show without a doubt that water does readily enter the CNT channels in a wide pressure range, including ambient conditions. This issue will be discussed in detail in Section 4.

Upon water entering the CNTs, one can argue that all its properties deviate from the bulk mainly due to the hydrophobic interactions with the carbon atoms and the effect of nanoconfinement. As it is already mentioned, the CNT walls made of pure graphene (i.e., with negligible defects and impurities) exert hydrophobic forces on the water molecules, which generally cause the formation of a depletion layer close to the walls [74], a number of concentric water tubes towards the center of the CNT, plus possibly a chain of stratified water molecules at the very center, depending on the diameter of the CNT [90]. Such effects are detailed in Section 5. Additionally, the presence of the hydrophobic wall and the restricted available space change the structure of the hydrogen-bond network of the water molecules, reducing significantly the average number of bonds per molecule compared to its bulk value. Section 6 discusses how this reduction affects the properties of nanoconfined water.

Perhaps the most important issue in regard to water in CNTs is defining the properties of its diffusion. It has been shown both theoretically [4,75] and experimentally [63,64] that under certain conditions the dimensional restriction and the water-wall interactions cause an enhanced diffusion coefficient for water in CNTs, compared to bulk water. In addition, if the CNT is narrow enough (d < 1 nm), then water molecules cannot surpass each other any more, turning the dynamic process from the typical classical/Fickian diffusion to a single-file diffusion. In wider CNTs, water is organized in concentric rings, each with its individual diffusion coefficient and properties, yielding the important question of whether there is a CNT diameter that maximizes the water flow, as this could be very useful for a number of applications, such as the water purification process (vide supra). More details on these issues are presented in Section 7.

Another important research question in this field is related to the freezing temperature and other phase transitions of water in CNTs. Several studies have found that confined water remains in the liquid state well below the freezing temperature of bulk water ( 273 K). It seems that the concentric water rings freeze at around 240 K [11], although the central stratified water chain is shown to remain liquid at much lower temperatures ( 220 K) [52]. In any case, the actual freezing temperatures of these components depend on the CNT’s diameter and possibly the hydrogen isotope of water (i.e., heavy versus light water). These effects are discussed in Section 8, while Section 9 provides an outline of how external EM fields, impurities, defects, functionalization and surface roughness affect the properties of the CNTs. Finally, Section 10 summarizes all open (and contested) questions in this field and suggests avenues for future research.

## 2. Carbon Nanotubes: Synthesis and Characteristics

Iijima [91] is accredited with the discovery of CNTs, being the first to describe MWCNTs in 1991, while studying with electron microscopy an arc evaporation test during the fabrication process of Buckminsterfullerene (i.e., C${}_{60}$ or “buckyball”). Nonetheless, CNTs were to some lesser extend known much earlier, having been detected by Radushkevish and Lukyanovich [92] in 1952 and (SWCNTs) by Oberin [93] in 1976.

### 2.1. Synthesis

For the synthesis of SWCNTs, several methods have been utilized over the previous three decades [94]. The earliest method was arc discharge of carbon (graphite), typically in the presence of some metal catalyst, such as Fe, Co or Ni, [95]. The CNTs that are produced this way are generally closed at either end, but can be readily opened by contact with various oxidants, which also remove contaminants such as residual amorphous carbon and catalyst particles [95]. Other methods that have been surveyed for the synthesis of CNTs [96] include pyrolysis of precursor molecules, laser ablation, electrochemical synthesis and chemical vapor deposition (CVD). In CVD, some typical carbon-based raw materials include CO, acetylene, metallocenes and Fe(CO)${}_{5}$/C${}_{2}$H${}_{2}$ [95].

There are two types of carbon nanotubes that have so far been synthesized, each with its distinct properties. These are known as true carbon nanotubes and carbon nanopipes [97]. The former were first surveyed by Iijima [91] and have the cylindrical fullerene structure, while the latter have the honeycomb structure and are produced from CVD of carbon in alumina. The carbon nanopipe walls are typically made of amorphous carbon, in contrast to the CNTs, the walls of which exhibit the well defined structure of graphene. Unless stated otherwise, in this review we will reserve the acronym “CNT” to only refer to “true” carbon nanotubes.

### 2.2. Properties of CNTs

CNT molecules are made of sp${}^{2}$ hybridized carbon atoms on a bent hexagonally-arranged (graphene) lattice (see Figure 1). Based on the angle between the graphene unit cell and the longitudinal coordinate of the tube (or in other words based on the chiral—or Hamada—vectors of the CNT’s lattice), one can identify three types of CNTs; the chiral, the armchair and the zigzag. The relevant Hamada vector of the CNT is usually written as (n, m). Within this nomenclature, armchair CNTs have chiral vectors of the form (n, n) and are always metallic, the zigzag CNTs are noted as (n, 0) and the chiral tubes have arbitrary vectors (n, m). The latter two types are metallic (or very narrow-gap semiconductors), if (n-m) mod 3 = 0, else they are semiconductors [45] (see Figure 1a).

The distance between a given carbon atom and its three nearest-neighbors is roughly 1.42 Å, with a σ-bond (resulting from the overlap of their sp${}^{2}$ hybrid orbitals) connecting the nearest-neighbors, as is also the case for graphene and fullerene [99]. As was mentioned in Section 1, CNTs are identified in terms of the number of graphene sheets that comprise their walls as single-walled (SWCNTs), double-walled (DWCNTs), or multi-walled carbon nanotubes (MWCNTs). In DWCNTs and MWCNTs, the interlayer distance of adjacent graphene sheets is roughly 3.35 Å, each sheet forming a concentric tube that is coupled to its neighbor(s) by a π-bond, as do consecutive layers in graphite. SWCNTs, having a simpler structure, usually have better-defined walls and fewer structural defects than MWCNTs. It is evident that DWCNTs and MWCNTs have an inner and an outer diameter (termed ID and OD, respectively), while in the case of SWCNTs there is no such distinction. In this review, unless stated otherwise, the term “diameter” refers to the internal width of the CNT, as this is the relevant dimension when it comes to confined water.

## 3. Methods for Studying the Properties of Water in CNTs

Although water properties inside CNTs have been studied with a plethora of experimental and theoretical/computational techniques, as explained in Section 1 in this review we will focus mostly on studies utilizing nuclear magnetic resonance and molecular dynamics. For this reason, it is useful to provide a short background on both of these important techniques.

### 3.1. Nuclear Magnetic Resonance

Nuclear magnetic resonance (NMR) is a well established, non-destructive, robust technique used widely in chemistry and condensed matter physics to extract information on the local (electro)magnetic environment of NMR-active (i.e., non-zero spin) nuclei in a sample. NMR has several advantageous features that explain its wide applicability and usage. Importantly for the case of water diffusing in CNTs, NMR has been demonstrated to be a powerful experimental tool capable of extracting the diffusion coefficient of water molecules [58,63,64,77], of distinguishing various water groups (e.g., between stratified, interstitial and bulk water) based on their dynamics and diffusion properties [52,63,64] as well as of identifying the freezing temperature of the various water components [49,52,53,58,60,63,64]. Typically, NMR studies processes on the microsecond timescale, i.e., roughly three orders of magnitude slower than the pico-to-nanosecond timescale of MD simulations. As a result, NMR provides valuable information on slower processes and steady-state phenomena [63], which are typically inaccessible using computational techniques.

The physical basis of NMR is the Zeeman interaction between the non-zero nuclear spins of specific isotopes in the sample and magnetic fields, both internal and external. In more detail, during an NMR experiment the sample is generally placed inside a strong, static, magnetic field B${}_{0}$, typically on the order of several Tesla. In this field, the nuclear probes’ longitudinal spin polarization acquires a steady-state distribution, which is perturbed by applying a much weaker oscillating radio frequency (RF) field B${}_{1}$. Upon removing the RF field, the nuclear spins start precessing and thus induce a signal on a pick-up coil, with the frequency of the induced signal characteristic of the *total* field at the site of the nuclei, consisting of *both* the extrinsic field B${}_{0}$, as well as the intrinsic magnetic environment around the nuclei. This way, the magnetic environment (both static and dynamic) in the vicinity of the NMR-active nuclei can be studied with a very high degree of accuracy.

To perform NMR experiments, the sample is placed in the bore of a—typically superconducting—magnet, with the sample’s temperature controlled usually using a combination of resistor heating and liquid N${}_{2}$ (or liquid He for temperatures lower than ∼200 K). Around the sample there is a coil connected to a resonance circuit, which both produces the RF pulses and (usually) picks-up the NMR induction signal. Since the NMR signal is typically orders of magnitude weaker than the applied RF, dedicated electronics (filters and amplifiers) should be used for signal acquisition.

For the simple case of I=1/2, the nucleus can be thought as an electric monopole and a magnetic dipole (i.e., it has charge +Z and a magnetic moment μ), but for I>1/2 there is also a non-zero electric quadrupole moment eQ, due to the fact that the distribution of charge for high-spin nuclei is non-spherical. As a result, nuclei with spin I≥1 interact with an electric field gradient (EFG) present at the nuclear position. This effect can be used to study, amongst other things, phase transitions of solids, self-diffusion, etc. For the I>1/2 case, one can distinguish between the effects associated with magnetic and electric interactions either by varying the static magnetic field $B_{0}$ (which would affect the magnetic—but not the electric interactions), or by comparing the NMR signal under identical conditions to that of a second NMR-active isotope of the same element [53,100], as different isotopes generally have different spin and/or quadrupole moments, so their couplings to these fields would be scaled accordingly.

The most commonly used isotopes in NMR spectroscopy are ${}^{1}$H, ${}^{2}$H and ${}^{13}$C [101]. In the case of water in CNTs, by far the most common probe is ${}^{1}$H, followed by ${}^{2}$H (i.e., heavy water) [53,60]. Heavy water is studied in this context with NMR because it exhibits a distinct NMR signal compared to regular (${}^{1}$H${}_{2}$O) water, especially if the molecules are restricted in their motions where the NMR power-pattern lineshape might be obtained. This signal gives quantitative information about the mobility of the water molecules on the surface of CNTs.

The main limitation for a wider use of ${}^{13}$C NMR is its inherently poor signal-to-noise ratio (SNR). Indeed, SNR scales linearly with the number of NMR-active nuclei in a sample, as well as with their gyromagnetic ratio raised to the 5/2-th power (i.e., SNR $\propto \gamma^{5/2}$). Both these factors are significantly suppressed for ${}^{13}$C, compared to ${}^{1}$H. In a sample containing carbon, only 1.1% of carbon nuclei are ${}^{13}$C (with non-zero spin) and the rest are ${}^{12}$C, plus possibly some tracer concentration of the metastable ${}^{14}$C, both of which are zero-spin nuclei. Furthermore, the value of γ for carbon-13 is about one quarter that of ${}^{1}$H. Thus, the a priori SNR of ${}^{13}$C NMR is roughly 30 times smaller than that using protons as probes [101].

There are several important NMR methods, many of which can and have been used to extract complementary information on water in CNTs. Historically, the first NMR method to be developed is that of *lineshape* spectroscopy, with which one can study the quasi-static (or time-averaged) parameters of the spin Hamiltonian (e.g., local magnetic field or electric field gradient). In this mode, the NMR signal is plotted in frequency space (in modern experiments by using pulsed RF fields and performing a Fourier transform of the resulting signal). This generally produces a number of resonances (peaks) when the frequency of $B_{1}$ matches the Larmor resonance frequency $\omega_{L}=\gamma B$, where *B* is the sum of the applied field $B_{0}$ *and* the local magnetic field at the site of the nucleus. Thus, each resonance peak reflects the existence of an inequivalent environment (site) that some of the probing nuclei occupy, with the relative intensity of each peak being proportional to the percentage of spin-probes in that (chemical) environment—assuming that the effects of nuclear Overhauser and cross-polarization are negligible.

The position of the peaks (and their possible variation with temperature) can provide information on static interactions and magnetic (or more generally phase) transitions, while the width of the peaks (typically expressed as full width at half maximum, FWHM) informs of the (near-resonance) dynamical interactions between the nuclear probes and their environment. The resonance shift is also called the *chemical* shift, if it stems from interactions of the nuclear spins with the orbital motion of the neighboring electrons, or the *Knight* shift, if it is due to the presence of unpaired electron spins, e.g., in metals. In any case, the shift of the peak position—compared to that of a reference material—allows for the study of the local internal magnetic field. Essentially, the chemical shift measures how much of the external magnetic field’s strength is shielded at the site of the nucleus by its surrounding electron cloud [101]. Tetramethylsilane (TMS) is commonly used as reference, with the position of the resonances in other materials expressed in parts per million (ppm) relative to the narrow peak of TMS.

Information on the fluctuations of the electromagnetic fields in a sample is obtained through measurements of the (longitudinal) spin-lattice relaxation ($1/T_{1}$) and the (transverse) spin–spin relaxation ($1/T_{2}$) rates. The relaxation rates are extracted using pulsed RF sequences (e.g., inversion and saturation recovery for $1/T_{1}$, Hahn and Carr-Purcell-Meiboom-Gill—CPMG, for $1/T_{2}$). The measured relaxation rates depend mostly on those fluctuations that have an appreciable spectral density close to the Larmor frequency. Since $\omega_{L}$ is related to the *total* magnetic field, the relaxation rate depends on the applied magnetic field $B_{0}$, so the latter can be varied (e.g., using a second superconducting magnet of different strength) to allow for the study of fluctuations at different frequencies.

Direct information on diffusion processes can be extracted using pulsed field gradient (PFG) [61] and stray field gradient (SFG) [64] NMR. In the first case, a short spatially-varied pulse is used, whereas in the latter case the sample is placed in the stray field of the magnet, in the region where the field’s intensity changes linearly with position. Both techniques create non-equivalent (magnetic) environments, thus permitting the study of the motion of spins through them. Additionally, NMR can study diffusion indirectly, based on the effective fluctuating fields felt by the nuclear probes due to their motion. Fast molecular motion causes dynamical averaging of the fluctuating fields felt by the spin-probes, thus leading to motional narrowing of the NMR peak (i.e., reduction of the corresponding FWHM with increasing temperature). Furthermore, when plotting the relaxation rate versus temperature ($1/T_{1}$ versus *T*), a peak appears at the temperature where the frequency of the fluctuating fields produced by the diffusing motion of the nuclear probes matches the Larmor frequency [102]. 1D nanoscale diffusion has also been studied in several systems using more “exotic” NMR-related techniques, such as β-detected NMR (β-NMR) [102] and muon spin rotation-relaxation (μSR) [103], which detect the NMR signal using extrinsically implanted spin-probes, utilizing the fact that the Weak nuclear force is not parity-symmetric (i.e., the fact that the direction of the emitted β-particle is correlated with the direction of the nuclear—or muonic—spin at the time of the decay) [104].

Often, the diffusivity, *D*, and the relaxation rates $1/T_{1}$ and $1/T_{2}$ do not acquire a single value in a sample at a given temperature, but they come as a distribution, especially if there is a continuum of non-equivalent local environments felt by the spin-probes. In such instances, the normalized NMR signal (decay curve), g(t)/g(0), is modeled using a Fredholm integral of the first kind:(1)$$\frac{g(t)}{g(0)}=\int_{0}^{+\infty }k_{0}\left(x,t\right)f\left(x\right)d\left(log_{10}x\right),$$
where *x* = $T_{1}$, $T_{2}$ or *D*, f(x) is the distribution in question and $k_{0}\left(x,t\right)$ is the kernel function of the signal decay (e.g., $k_{0}\left(T_{1},t\right)=exp\left(-t/T_{1}\right)$). By inverting the above equation, one can solve for the distribution function f(x) [105].

Although the aforementioned NMR methods (lineshape, relaxation and diffusion analyses) have provided extremely valuable information on water confined in CNTs and helped to determine the existence of multiple water groups with different dynamics, they cannot distinguish the correlations between these groups, nor can they study them individually. This can be proven problematic, given that MD simulations reveal the existence of water components with decisively different dynamics and diffusive properties (stratified water chain at the center of the CNT, coaxial water rings, etc., see Section 5). If the distributions of their diffusivities partially overlap, the above usual NMR techniques would only yield the *total* distribution of *all* water components taken collectively, potentially obscuring details of the various water components. To individually study each water group and its different diffusive properties, one can perform 2D NMR experiments, in which *two* rates are studied at the same time (i.e., $T_{1}$-$T_{2}$, or *D*-$T_{2}$ NMR), yielding robust information of correlations between the dynamics and the diffusivity of each water group [63,64]. Thus, this way one can untangle the distribution functions of the diffusivity of *each* water component based on their relaxation rates (see Figure 2). This is further discussed in Section 7.

Utilizing the above strengths and complementary methods, NMR studies of water in CNTs were able to yield significant results, hard-to-get with other techniques, especially experimental ones. Indeed, NMR was shown to be able to distinguish between external (bulk), interstitial and internal water, based on their different dynamics [52,63,64] and in systems that support multiple distinct water components inside the nanotubes (see Section 5), NMR studies were able to probe these components *individually*, in terms of water diffusion properties [58,63,64] and ice structure [53]. Further, NMR studies identified the *type* of motion of the water molecules (Fickian, single-file, etc., see Section 7) and extracted the diffusion rate and its temperature dependence, which was then used to identify the fragility of nanoconfined water [63,64]. In addition, a large number of NMR studies utilized lineshape analysis in order to study the water-ice phase transition inside CNTs of various diameters [49,52,53,58,60]. They found that internal water retains its liquid character well below the freezing temperature of its bulk counterpart ( 273 K), with a central water-chain remaining in liquid state as low as 220 K. Finally, based on the resonance peak’s shift (relevant to TMS, see above), the chemical shielding of the magnetic field at the nuclear position is identified, caused apparently by ring currents in MWCNTs that are stronger than in their SW and DWCNTs counterparts [52,56,57,62].

### 3.2. Molecular Dynamics Simulations

With the rise of modern computers and their exponentially increasing processing power, it has become ever easier to utilize them in order to extract the physical characteristics and to study dynamical processes of microscopic systems. In the context of chemistry and condensed matter physics, the most powerful computational tools in this manner are molecular dynamics (MD) [4,22,30,65,66,67,68,69,70,71,72,73,74,75,76,77,78,79,80,81,82,83,84,85,86,87,88], density functional theory (DFT) [68,106] and ab initio MD (AIMD) [67,70]. For the system at hand (water in CNTs), the best-suited computational techniques are MD and AIMD simulations, as they can study diffusive processes over nanoseconds of simulation time and are amenable for simulations of large systems.

MD simulation in particular is a powerful theoretical tool that can provide insights at the atomic level, which would be nearly impossible to extract experimentally. Based on these results, one could then offer quantified predictions to be verified experimentally, or explain already-acquired experimental data. In contrast to DFT, which is very computationally-intensive when surveying large systems, MD simulations can promptly study systems comprising of a large number of interacting particles. Each particle can be modeled with a varying level of detail, with well-refined potentials governing the important interactions. For example, the water-carbon van der Waals interactions, governing the characteristics of water in CNTs, are typically modeled using Lennard-Jones (LJ) potentials. The water-water interaction is modeled in the literature using one of several possibilities, such as the simple point-charge (SPC), the flexible SPC, the simple point-charge extended (SPC/E), as well as three-site (TIP3P), four-site (TIP4P) and five-site (TIP5P) water models [29,75]. For example, the SPC/E model represents each water molecule as a sphere with the oxygen atom at its center and partial charges at the oxygen and hydrogen sites [107]. Turning to the structure of the CNT wall, note that most MD simulations of wetted CNTs used a rigid-wall model, in which the carbon wall is fixed. A few studies chose a more flexible wall model, such as Associated Model Building with Energy Refinement (AMBER) and Chemistry at HARvard Molecular parameters (CHARM) [75].

From the above, it is clear that a crucial aspect of MD simulations relates to choosing good models for the forces at play. In classical MD these forces are often computed using empirical models, derived for bulk water. This raises questions on the validity of the associated water-model in nanoconfined geometry, especially since even the most commonly used and widely tested water-models approximate well only certain properties of (bulk) water [75]. In addition, in this confined nanoscale geometry, it is not clear a priori if it is prudent to neglect relevant quantum effects [36,37,108]. Similar concerns have been raised about the common assumption of a rigid CNT. According to the work of Jakobtorweihen et al. [109] on CH${}_{4}$ confined in CNTs, the (in)flexibility of the tube’s wall impacted the physical properties—and especially the diffusivity—of CH${}_{4}$ at low pressure (*p* < 0.05 bar), but had negligible influence at higher pressure. In recent years, the task of choosing proper MD models is made more rigorous with the development of AIMD, which computes the forces using directly the molecules’ electronic structure [70].

After the initial geometry of the system is defined and the specific model for interactions has been selected, the system is allowed to evolve under the influence of Newtonian Mechanics. Using this methodology, MD simulation allows the extraction of both dynamical and equilibrium thermodynamic properties at any finite temperature. This is in contrast to DFT, which studies the ground state of the system (at “zero Kelvin”), but on the other hand takes quantum effects into account.

## 4. Intercalation of Water in CNTs

As was mentioned in Section 1, CNTs can readily be wetted, in spite of the hydrophobic nature of their walls. This result was counterintuitive at the time, also because spatial confinement generally is expected to decrease the entropy of the wetting process [110]. Nonetheless, the intercalation of water in CNTs was established as a fact by both experimental and computational studies. The former include a large body of evidence, using, amongst others, neutron scattering [111], optical [112,113] and X-ray [114] measurements, as well as NMR spectroscopy [49,52,53,58,60,63,64].

Turning first to the theoretical studies (i.e., MD simulations), already since 2001, Hummer et al. [65] were able to show that water fills narrow (d < 1 nm) empty SWCNTs in bursts, which apparently stem from the tight hydrogen-bond network in the CNTs. The stochastic nature of the flow is attributed to the fact that at the nanoscale, thermal fluctuations become important [4]. Apart from the aforementioned temporal variation of water intercalation, there is also a spatial component to the wetting process of CNTs. The hydrophobic nature of the CNT wall causes water to first fill the center of the tube and only then (as the water density increases) it starts filling the near-wall region [115]. This is the exact opposite to the filling process of water in hydrophilic materials, such as MCM-41 [116,117].

Subsequent MD simulation studies established that the choice of the intermolecular potential is crucial as to whether the CNT will be wetted or not in a simulation [66,97]. This sensitivity to the selected potential might stem from the fact that the water molecules lose roughly two of their four hydrogen bonds upon entering narrow CNTs (see Section 6) and they recover only a small fraction of that energy through their van der Waals interactions with the CNT’s walls (see Section 5). As a result, the Lennard-Jones potential well, which is what allows the wetting of the CNTs to happen, is very shallow (0.114 kcal/mol) [65] in narrow CNTs—and thus very sensitive to the choice of the potential. Note, however, that the more recent detailed simulations of Pascal et al. [110] (see below) predict a much deeper potential for the narrow CNTs, as low as −4 kcal/mol in CNTs of 8 Å diameter.

Wetting is also pressure and temperature dependent. Ohba et al. [118] found that at 303 K, below a critical pressure of $p=p_{0}/2$, where $p_{0}$ is the saturated vapor pressure of water ($p_{0}$ = 4.2 kPa), water will not get uptaken in CNTs. Furthermore, using a combination of NMR, XRD and electrical resistance measurements, Kyakuno et al. [60] reported a wet-dry transition in SWCNTs with diameters between 1.68 and 2.4 nm below a critical temperature ranging between 218 and 237 K, with the phase transition increasing with increasing CNT width (see Section 8).

### The Thermodynamical Perspective of Wetting CNTs

To further examine the above phenomena, including the effect of the nanoconfinement on the wetting process, CNT systems have been studied from a thermodynamic perspective, whereby the water uptake of the CNTs is considered in terms of the process’ energetic and entropic terms. In the literature, the most common approaches [119] for estimating the free energy of wetting using MD simulations include the calculation of Potentials of Mean of Force (PMF) [72], particle insertion strategies [65,120] and analysis of the water’s velocity autocorrelation function [110]. Interestingly, even in small CNT diameters (d < 1 nm), in which water forms a single-file arrangement (see Section 7), water molecules have considerable entropy because they freely rotate about their hydrogen-bond-chain [65].

A systematic study of both energetic and entropic terms of the free energy of water inside CNTs versus the width of the tube was conducted by Pascal et al. [110] (see Figure 3). For all CNT diameters they studied (between 0.82 and 2.72 nm), the free energy of water inside the CNT is lower than that of bulk water, with local minima at the 0.8 and 1.2 nm widths. This means that water is expected to readily flow into the CNTs, in agreement with experimental results. Nonetheless, the *mechanism* permitting the water filling seems to heavily depend on the width of the tube. According to Pascal et al., water intercalation in narrow (0.8– 1 nm) CNTs is entropy-stabilized (with both rotational and translational components) and generates a vapor-like phase. In CNTs of medium width (1.1– 1.2 nm), enthalpy stabilizes the process and creates an ice-like phase, while for wider CNTs (above 1.4 nm diameter), wetting is stabilized by the increased translational entropy (due to the wider available space) and forms a bulk-like liquid phase. This is in agreement with the MD simulation of Gauden et al. [121], who found a strong maximum of the enthalpy of the uptake process at a CNT’s diameter of 1.17 nm. Further, Pascal et al. compared their thermodynamic results with MD simulations using simpler water-models, such as AMBER95, single particle M3B and monotonic water model (mW). Notably, with the M3B model water would not enter the CNTs, regardless of their diameter, whereas mW’s entropy profile is compatible with that extracted with the SPC/E model, except for CNTs of subnanometer width, in which both AMBER95 and mW models underestimate the entropy of the wetting process.

For narrow SWCNTs, such as armchair (5, 5) and (6, 6), several MD studies arrived at somewhat contradictory results. For instance, Waghe et al. [120] simulated (6, 6) CNTs between 280K–320K and reported a positive energy but negative entropy for the water transfer process. This means that water uptake is an energy-driven process and the counter-effect of entropy would cause the CNTs to empty at higher temperatures. Gerate et al. [119], studied the thermodynamics of wetting both (5, 5) and (6, 6) CNTs and found that water uptake of the (5, 5) is not thermodynamically favorable, whereas (6, 6) CNTs are favored by both energy and entropy terms. Their conclusion for (6, 6) CNTs was also supported by Kumar et al. [122], who also reported that with increasing temperature both translational and rotational entropy decreases, as does the energy transfer.

In conclusion, water filling of CNTs is a process governed by both energetic and entropic effects, with their relative significance strongly depended on the width of the tube. Nonetheless, the specific properties of the process seem to depend on the parameters and models used for the MD simulations, yielding some contradicting results at the narrow end of CNT diameters [85].

## 5. Why Water in CNTs Does Not Behave like Bulk: Nanoconfinement and Hydrophobicity of the CNTs

### 5.1. Effects Induced by Nanoconfinement and Water-Wall Interactions

The geometry of the CNT has a huge impact on the properties of intercalated water. For instance, water flow in CNTs has been shown experimentally [123,124] to be between two and five *orders of magnitude* faster than what flow theory (Hagen-Poiseuille equation) predicts (see Section 7.4). To understand the mechanisms governing the dynamics of water in CNTs, a large number of MD simulation studies have been employed [32,75,125]. This exceptional flow is attributed to the “smoothness” of the CNT walls [97,126], which increases the water velocity close to the wall by creating a favorable water orientation and hydrogen bonding [74]. In the depletion region close to the wall, water molecules form dangling hydrogen bonds which, in turn, increases the degrees of freedom of near-wall water molecules and aids their diffusivity [127]. In other words, the hydrophobic nature of the CNT wall essentially makes the CNTs to act as frictionless pipes [128], which greatly enhances water dynamics.

This enhanced flow critically depends on the water-wall interactions [129], which was excellently demonstrated by Merillo et al. [130] using a series of MD simulations with varying water-wall interaction strength and CNT’s diameter. Based on these simulations, when one varies the strength of the oxygen-carbon interaction, there is a narrow transition region (between 0.05 kcal/mol and 0.075 kcal/mol), in which the change of water flow and occupancy in CNTs steeply increases with increasing interaction strength. The specific onset of this process depends evidently on the diameter and length of the CNT channels. The transition region of interaction strength coincides with forming nearly vertical water-wall contacts, which seems to indicate that this is where the connection of the water-wall interactions and its effects on diffusion stem. According to the MD simulation of Mukherjee et al. [77], the effect of nanoconfinement on the orientational degrees of freedom is uneven. On the one hand, the orientational relaxation time of the average dipole moment is longer by 3 orders of magnitude, compared to bulk, while the relaxation of the H-H vector inside a nanoconfined water molecule is roughly ten times faster than bulk. Nanoconfinement might also cause phase transitions of water into phases not accessible with bulk water [65], which will be discussed in further detail in Section 8. Additionally, water molecules are predicted to couple to the graphene wall’s longitudinal phonons [131], which could enhance their diffusivity by more than 300%, in a motion resembling that of a surfer catching a wave [131].

It is evident that all effects connected to either nanoconfinement or water-wall interactions should greatly depend on the width of the CNT. Borg et al. [90] excellently showed this using MD simulations (Figure 4). Indeed, Figure 4 will act as the “Rosetta’s stone” for the rest of the review, as it shows the different water-structures that are present at various CNT diameters. Thereon, we will be using these distinct diameter-ranges to understand results from the literature that *seemingly* contradict each other.

### 5.2. Structure of Water in CNTs

Initially, the upper limit of the CNT’s width for water in it to be considered confined was debatable, with reported values as wide as 100 nm [132], to as narrow as 1 nm [73,97]. Nonetheless, a wide range of studies (both experimental and theoretical, see below) showed that there is a gradual shift from extreme confinement in narrow CNTs towards bulk-like water in wider tubes. Based on the density profiles shown in Figure 4, here we will identify four *distinct* diameter-ranges, each allowing different water-structures in the CNTs (see below).

#### 5.2.1. Water in Sub-Nanometer CNTs: Central Water-Chain Moving in Single-File

For narrow CNTs (with diameter less than 1 nm), water adopts a single-file chain structure, with characteristics drastically different than bulk water [4,65,133] (see Figure 4a,b). This was already evident in Hummer’s 2001 study [65], who reported that in narrow CNTs, the single-file water chain forms a 1D hydrogen network, having bonds stronger than those in bulk water. This water-chain is found to be highly ferroelectric [134], a result that could prove very important for future applications. Karla et al. [4] studied CNTs with a diameter of 8.1 Å using an intricate geometry whereby the CNTs connected a water and a salt reservoir, a scenario relevant to biomembranes. They reported that individual water molecules were diffusing in a single file (see Section 7), but the water chain can be thought as moving collectively by means of a 1D random walk. In a recent study, Tunuguntla et al. [133] used a MD simulation to study the water permeability of an 8 Å-wide CNT, also reporting the formation of a single-file water chain. They found that water flow in narrow CNTs is larger, by an order of magnitude, compared to that in wider CNTs and that in biological membranes. In addition, they showed that ion transfer through narrow CNTs can be tuned by switchable diodes, making narrow CNTs promising materials for desalination projects.

#### 5.2.2. Water in CNTs of Diameter between 1 nm and 1.2 nm: Tubular Water Structures

In CNTs of larger diameters ( 1.0 nm < d < 1.2 nm, see Figure 4c,d), water’s structure becomes much more interesting. In an early MD study, Koga et al. [135] found that in this range of diameters, water forms ice-like nanotubes, i.e., rolled sheets of water molecules, with their geometrical structure (square, pentagonal, hexagonal, and heptagonal) depending on the thermodynamical conditions such as temperature and pressure. By combining MD simulations with X-ray diffraction (XRD) measurements, Ohba [136] found that ice-like tubular structures appear for diameters above 1.0 nm and melt in larger sizes above 3.0 nm. This intermediate-diameter range has been studied extensively with several experimental techniques since the early 2000’s, including XRD [137,138], neutron diffraction [111] and NMR [49,53,58,60], all of which verified the aforementioned theoretical predictions. The results of these studies regarding the ice-structures and water-ice transition temperature of confined water are further discussed in Section 8.

Due to the nanoconfinement effect on water-water and water-wall collisions, water molecules in narrow CNT sizes of ∼ 1.0 nm are found to exhibit a fast NMR relaxation of 1 ms, which gradually slows to a value of 440 ms, comparable to that in bulk, when CNT sizes increases to 2.0 nm [62]. The enhanced relaxation of water in narrow CNTs stems (according to MD simulations [62]) from the rapid energy transfer between water molecules and the CNT wall, whereas at the wider (d ∼ 2 nm) tubes, the relaxation is caused mostly by the water-water collisions, which are limited due to the cluster formation of the water molecules.

#### 5.2.3. CNT Diameters between 1.2 and 
3.5 nm: Co-Existence of Central Water-Chain, Surrounded by Several Water-Tubes

In the diameter range (1.2 < d < 3.5 nm), things become even more intricate (see Figure 4e–i). In this region, the CNT is host to *distinct* groups of water, each with different characteristics. Indeed, MD simulations [74,139,140] predicted that at the very center of these CNTs a chain-like network of water molecules develops, exhibiting fast diffusivity and stratified collective motion. Around that chain, one (or several, depending on the CNT’s diameter) water-tube(s) are formed, first as far away from the wall as possible (due to the associated hydrophobic water-wall forces), whereas in wider CNTs the water-tubes start to become increasingly less rigidly organized and start resembling bulk water. This arrangement was verified by early neutron scattering experiments [111] and the NMR (lineshape) 2004 study of Gosh et al. [49], who studied CNTs of 1.2 and 1.4 nm diameter. Nonetheless, two recent NMR studies were able to collect much more detailed information on the dynamics and diffusion properties of these water components [63,64]. The 2018 study of Hassan et al. [63] studied 1.2 nm SWCNTs and 3.5 nm DWCNTs using a combination of 2D NMR and MD simulations. Utilizing T${}_{1}$-T${}_{2}$ and D-T${}_{2}$ NMR spectroscopy (see Section 3.1), they found that in the narrow SWCNT there is only one water group—that of a single tube of water close to the center of the CNT—in agreement with other studies of this diameter region, but, interestingly, in the wider DWCNTs, they found both a central water-chain of stratified water and a surrounding water-tube. The diffusion profiles of these two water-groups were partially overlapping, but using the power of 2D NMR, they were individually resolved by means of their distinctive dynamics: in the 2D D-T${}_{2}$ spectrum, the two groups’ different relaxation profiles were used to untangle their respective diffusion profiles.

#### 5.2.4. Above CNT Widths of 4 nm: Internal Water Approaches Its Bulk Properties

Upon increasing the CNT sizes to above ∼4 nm, the confinement effects and water-wall interactions become progressively less pronounced (Figure 4j). Therefore, both the average number of hydrogen-bonds per molecule (see Section 6) and the internal water’s diffusion rate (see Section 7), gradually approach their values for bulk water [127]. The same was found to be true for the NMR relaxation rate [62].

The gradual evolution of internal water’s structure with the CNT’s width was demonstrated recently by Gkoura et al. [64], using a combination of 2D NMR and MD simulations (see Figure 5). They studied systematically the properties of internal water in CNTs with diameters ranging from 1.1nm–6.0nm. They reported that for the relatively narrow CNTs ( 1.1 nm), water forms a single water-tube close to the center of the CNT (in agreement with Figure 4d), whereas at the ∼3 nm range there is a stratified central water chain engulfed by a water tube (similar structure as that found in the DWCNTs of comparable diameter in Ref. [63]). Interestingly, they were able to resolve the overlapping diffusion profiles of the central ring and surrounding water-tubes based on their different dynamics (i.e., using the fact that these groups exhibited different T${}_{2}$ relaxation profiles) and showed that the central chain forms at a width above 1.1 nm and gradually dissolves above 4 nm, becoming completely negligible for widths wider than 5 nm. Above that width, the diffusive properties of water readily approach that of bulk water, as was predicted by earlier MD studies [127].

In contrast to the great importance of the CNT’s diameter on the water’s parameters, the chirality of the tube (see Section 2) does not affect the water’s properties very much. Indeed, Wang et al. [141] found with MD simulations that the chirality of the CNT has very small influence on the properties of the nanoconfined water; but on the other hand, Tanghavi et al. [142] reported that zigzag SWCNTs allow a lower water diffusion coefficient, compared to SWCNTs of the armchair chirality. This result suggests that one should be careful when extrapolating the results of water intercalating in armchair CNTs to tubes of other chiralities, especially in regards to the diffusivity, which as will be discussed in Section 7 allows for the largest discrepancies between studies, even under very similar conditions.

## 6. Hydrogen Bond Network

The ultimate root of all observable characteristics of water in CNTs (its diffusive properties, phase transitions, etc.), which will be surveyed in Section 7, Section 8, Section 9 and Section 10, are the confinement and water-wall interactions discussed above. However, the *proximal* cause of the particular attributes of water in CNTs can be argued to be the structure of the hydrogen bonds connecting neighboring water molecules. Here we study how the effects discussed in Section 5 influence the H-bond (HB) network, giving the water under confinement characteristics very different from those in bulk water.

### 6.1. Pressure and Temperature Dependence of the HB-Network

According to a number of theoretical MD simulation and experimental studies, the structure of the HB network depends on the width of the CNTs and the applied pressure. The effect of pressure on the H-bond network is evident when comparing the MD studies of Ohba et al. [118], with that of Pascal et al. [110]. In the first case, Ohba et al. studied water inside CNTs under low pressure (from 0 to 4.2 kPa) in narrow ( 1 nm) and wider ( 2 nm) CNTs and found that the average number of HBs was just 0.6 for the former and 2 for the latter, both significantly different than the bulk value of ∼3.7 bonds per water molecule. In contrast, Pascal et al. studied the thermodynamical properties and the HB network of water in CNTs in a range of diameters between 8 Å and 2.7 nm at 1 atm and 300 K. Although they also found the average number of HBs to increase with increasing width, they reported an average of 1.7 HBs per molecule in the 8 Å CNTs, 2.7 HBs per molecule for the 1.0 nm case and 3.5 HBs/molecule—very close to the value for bulk water—already for CNTs of 1.6 nm diameter. This dependence of the hydrogen bonds to applied pressure can be understood by noting that small pressure leads to a smaller water density in the CNT. It is then perhaps not surprising that in the low-pressure regime the average number of HBs is lower than that under high pressure.

It is interesting to note that temperature does not seem to affect the HB network as much as pressure [88]. For instance, in an early MD study by Marti and Gordillo [143], it was found that the water molecules had less HBs than the bulk water in the whole range of simulated temperatures (298 to 500 K) for CNTs with diameters between 4.1 and 6.8 Å and a fixed water density of 1 g/cm${}^{3}$.

### 6.2. HB-Network’s Structure versus the CNT’s Diameter

From the previous discussion it is already evident that the number of HBs per water molecule increases with the diameter of the CNT, until it reaches the bulk value of ∼3.7 bonds per molecule for wide tubes. This result was confirmed by Barati and Aluru [127], who studied the *spatial* variation of the average HB per molecule inside wide CNTs. Interestingly, they found that in wide—(20, 20) and (30, 30)—CNTs, the water located near the center of the tube had 3.7 HBs per molecule, just as the bulk water, while near the wall that value was reduced to 2.1, an indication that water near the walls of the CNT never really behaves like bulk water. At the other extreme (for widths less than 5 Å), Mashl et al. [22] reported just one HB per molecule, which readily increased towards its bulk limit for wider CNTs ( 8.6 Å). Another type of spatial variation of the HB network is reported by Byl et al. [112]. They combined density functional theory (DFT) ab initio calculations with vibrational spectroscopy to show that in CNTs of intermediate diameter (between 1.08 and 1.49 nm), in which water forms tube-like structures such as stacked rings (see Section 5), the intra-ring HBs are bulk-like, whereas the inter-ring ones are weaker. The latter produce a distorted geometry with a distinct OH stretching mode, which lead to two different vibrational features.

### 6.3. Discrepancies between Studies of the HB-Network

The results reported by several theoretical studies on water’s HB-network are not without discrepancies. For instance, based on MD hybrid Monte Carlo simulations—accompanied by XRD measurements, Ohba et al. [62,136], reported the number of HBs per molecule to range from 1.2 to 2.3 in 1.0 nm and 5.0 nm CNTs, accordingly, acquiring a maximum value of 2.8 HB per molecule at a diameter of 2.0 nm. Notably, their reported value of HB for bulk water was significantly smaller than other studies, namely 2.2 instead of 3.7. This might be because Ohba et al. report *strong* HB-bonds, whereas most studies [110] define the existence of an HB if the oxygen-oxygen distance is less than 3.5 Å and at the same time the O-H-O angle is less than 30. Differences and variations among the reported values from MD simulation groups might also be attributed to the sensitivity of the system to the choice of the phenomenological force-field used [37], although the impact of some parameters such as TIP3P versus SPC/E water models or rigid CNTs versus flexible CNTs have been found to have negligible effects on the HB-network [75]. In any case, most MD simulations utilize a classical molecular liquid picture for water, which might not be strictly valid. According to the quantum calculations of Reiter et al. [36], the ground state of the valance electrons of nanoconfined water is significantly different than bulk water, which could make some simple water models that assume weak electrostatic interactions to be inaccurate. In a subsequent publication, the same group found that the quantum electron state of water molecules depends on both temperature and the width of the CNTs. They verified their conclusions using X-ray and neutron Compton scattering [37].

### 6.4. Effects Induced by the HB-Network’s Structure

The HB-network topology of water in CNTs is a key factor in determining a number of its characteristics. According to MD studies [3,118], the HB-network defines the diffusion mechanism, both in terms of its nature (Fickian, single-file, etc., see Section 7) and in terms of the flow speed [74,144]. Further, the reduction of the average number of HBs in nanoconfined water—compared to bulk—is shown to suppress the water-ice transition temperature [58,143,145], an effect that will be surveyed in detail in Section 8. Finally, the HB-network’s structure plays a significant role also in the filling process of water in CNTs. In Section 4, the study of Hummer et al. [65] was mentioned, which found that the fast uptake of water was due to the strong HBs of water inside the CNT. The study of Pascal et al. [110] also connected the thermodynamics of water filling with the HB-network. They reported a linear relationship between the average number of HBs and the enthalpy of water in the CNTs. In the light of the above discussion on the relationship of the average HBs and the width of the tube (Section 6.2), it is perhaps not surprising that Pascal et al. found that the wider the tube, the more HBs each molecule forms and the lower the enthalpy in the channel, making it thus easier for water to enter wider CNTs.

In the 2003 study of osmotic processes by Karla et al. [4], they found the water flow to depend mostly on particle entry and exit events. Nonetheless, the underlying mechanism is still unclear, although several studies since that time have identified the entry/exit events as important for water flow [83,87]. For instance, it is not fully understood if the H-network at the edges of the CNT could influence the internal H-network. [144] Only in 2017, Tunuguntla et al. [133] studied how the intermolecular HBs are a key parameter controlling the entry/exit energy barrier and they showed that by manipulating them, one could enhance the water flow. In this context, a recent MD study by Hou et al. [144] found that water flow from outside the CNT towards the interior greatly increases if a continuous HB-network is formed, which connects the internal water with the water molecules at the rim (i.e., near the exit) of the CNT. They reported that if such a unified network exists, the water flow can increase by two-fold, compared to the case that it is absent.

## 7. Water Diffusion in CNTs

Diffusion is possibly the most important property of water in CNTs, as it determines several macroscopic attributes, such as the fluid’s viscosity and the details of the water flow, the permeability of the carbon tubes and the heat transfer through the CNTs [22,127].

### 7.1. Diffusion: Short Outline

The process of diffusion was first studied in detail in 1855 by A. Fick [146], when he published his eponymous two laws, which state that diffusion is driven macroscopically by a concentration gradient. Microscopically, diffusion can be thought as the product of a random walk process, whereby the diffusive species moves from point to point through a series of collisions, each of which abruptly changes its momentum vector. If each such collision is uncorrelated to the previous ones, then Fick’s laws can be retrieved in the macroscopic limit [147]. Nonetheless, diffusion in narrow one-dimensional channels, such as inside CNTs, offers interesting variations from the above classical regime [148]. If the width of the channel is narrow enough—comparable to the size of the diffusive molecule—then the molecules are not able to pass each other, but rather they diffuse in a single-file fashion. At the other extreme, under specific circumstances the motion of the diffusing molecules might be coordinated, resulting in a special dynamical process denoted as “ballistic” diffusion.

Mathematically, the rate of diffusion is connected with the square of the displacement during a given time period by the general formula:(2)$${<|r\left(t\right)-r\left(0\right)|}^{2}>\;=ADt^{n},$$
where r(t) denotes the center-of-mass position at time *t*, *D* is the diffusivity (generally *D* is a tensor, but in 1D is just a scalar commonly noted as the “diffusion rate”), A=2d, where *d* is the dimensionality of the motion (i.e., A=2 for 1D diffusion) and the exponent *n* depends on the type of diffusion. In the classical Fickian case n=1, whereas n=0.5 for single-file motion and n=2 in the limit of ballistic diffusion [11,127,149].

Far from being constant, the diffusion rate, *D*, depends heavily on temperature. This can be easily understood by invoking a microscopic picture of diffusion, whereby the diffusing particle resides inside a potential well (e.g., a harmonic oscillator) and it attempts to hop to the next position by overcoming an energy barrier $E_{A}$ [150]. Utilizing Boltzmann’s statistics, the probability of a successful attempt is proportional to $exp(-E_{A}/k_{B}T)$, where $k_{B}$ is Boltzmann’s constant and *T* is the temperature in Kelvin. This is the root of the (macroscopic) *Arrhenius* relationship, which claims that diffusivity increases exponentially with temperature. Although the aforementioned Arrhenius law is commonly obeyed in a wide variety of systems, it is by no means universally valid. Departures from Arrhenius law can stem from, amongst others, a temperature-dependent entropy, a complicated microscopic diffusion mechanism, quantum mechanical tunneling through the barrier (applicable for particles lighter than lithium), or the existence of impurities and microstructures. If the activation energy is not constant with temperature, the diffusivity might follow a “sub-Arrhenius”, or “super-Arrhenius” relationship [151].

The above general discussion is valid in a very wide range of systems including solids, liquids and gases. In the case of liquid diffusion, the distinction between Arrhenius and non-Arrhenius diffusion is done using the concept of *fragility*. Liquids are categorized as “strong”, if they obey Arrhenius law, else they are noted as “fragile”. In this classification, bulk water is considered as fragile liquid, because its diffusivity deviates from the Arrhenius-exponential law at low temperatures. Recent NMR measurements on the temperature dependence of water’s diffusion rates in CNTs [64], also indicate that confined water exhibits high fragility, with its diffusivity following Speedy–Angell power-law [152] (see Figure 6). Especially for CNTs with diameters between 3.0 and 4.0 nm, the deviation from both Arrhenius *and* bulk-water is the most pronounced. Outside that CNT size region, confined water was found to be still fragile, but with a temperature-dependence of its diffusivity closely resembling that of bulk water. This picture was subsequently validated with the recent MD study of Srivastava et al. [88]. The special character of that intermediate diameter-region can be understood once again using Figure 4. According to that picture, in this region (3.0– 4.0 nm diameter) the water-chain structure with its unique dynamics is present, while it is absent at all other diameters surveyed in that study (i.e., at 1.1 nm and above 4 nm, see Figure 5).

### 7.2. Methods for Studying Diffusivity

There is a large number of methods capable of surveying various aspects of diffusion in different settings (e.g., microscopic or macroscopic diffusion, in tracer or high concentrations, etc.). Several NMR-related methods are used commonly for probing atomic-scale diffusion. These include pulsed field gradient NMR and stray field gradient NMR both of which can directly extract the diffusion rate from their measurements. Spin-lattice relaxation spectroscopy [102], also provides indirect information on the local hop-rate of the diffusing species (see Section 3.1). Other experimental methods capable of studying the process of diffusion include X-ray diffraction [114], inelastic neutron scattering [111] and the radiotracer [153,154]. From a theoretical standpoint, molecular dynamics simulations [135,145,149] and ab initio methods (e.g., density functional theory) [106] are commonly used in this regard. Note, though, that DFT does not directly calculate the dynamics (it yields the structure of the system at “zero” Kelvin). Thus, to study diffusion with DFT, one creates a static energy map of the most energetically-favorable path of the diffusive particle and then uses some model (e.g., Einstein–Smoluchowski’s law) to connect that microscopic picture to macroscopic diffusivity [102,106].

### 7.3. Diffusion of Water in CNTs

The subject of water diffusion in CNTs has amassed over the years a significant amount of studies, which at a first glance yielded contradictory results. Some studies claimed that water in CNTs exhibits faster-than-bulk diffusion [4,75] while others reported slower dynamics [22,79,118]. Some report a restricted water flow [22] and others a significantly fast flow rate [4,74,87,124]. To make sense of these (apparent) discrepancies, we have to remember the distinct diameter-ranges that we identified in Section 5, each of which has its own characteristics in terms of nanoconfinement, significance of water-wall interactions and topology of the H-bond network (Section 6). Thus, here we try to untangle the results in the literature in each of the following width ranges: (a) CNTs with diameters less than 1 nm, (b) 1 < d ≤ 1.2 nm, (c) 1.2 < d < 3.5 nm and (d) 3.5 < d ≤ 6 nm. Above a CNT width of roughly 6 nm, we can already say that water diffusion gradually approaches its bulk, unrestricted form. In each of the above ranges, we will survey what kind of motion is present (e.g., single-file, classical, etc.), how fast water diffuses in the CNTs and whether the flow is enhanced or restricted.

#### 7.3.1. Type of Motion

The first important question one needs to resolve in regards to water diffusion in CNTs is the type of motion. Namely whether diffusion is classical (Fickian), single-file or ballistic. It turns out that all three processes are present in this system, depending on the width of the CNT, the distance from the wall and the timescale of the measurement. When the CNT is empty and it is brought in contact with a water reservoir, MD simulations suggest that initially (i.e., during the first nanosecond), water pours into the CNT in a ballistic fashion [149,155,156,157], quite possibly in discontinuous bursts [65,158]. Unfortunately, this timescale (hundreds of picoseconds) is virtually inaccessible by experimental techniques, so only simulation studies can probe the very early dynamics of water diffusion.

Single-file diffusion is reported in narrow (d < 1.0 nm) CNTs by a very large number of studies, both theoretical [4,65,66,75,76,159] and experimental (using NMR [58,63,64]). It is worth mentioning, however, that in short CNTs, in which all water molecules might form a single, unified, water chain, their collective diffusion might be considered classical, even though individual water molecules cannot bypass one another [149]. It is thus noted by Foroutan et al. [29] that single-file diffusion is observed in narrow CNTs only if the geometrical and thermodynamic parameters generate multiple water-clusters in the tubes. Intriguingly, a very similar situation was reported by Taghavi et al. [142] using MD simulations of water inside single-walled silicon carbon nanotubes (SWSiCNTs). In SWSiCNTs of diameter less than 1.0 nm, they found an early ballistic diffusion mechanism evolving to single-file after the first half nanosecond, in striking resemblance to CNTs. This suggests that the effects of nanoconfinement on the properties of water are to a certain degree applicable to a wider range of nanotubular materials, although one has to be very careful not to assume that they are universally valid (see Section 9).

In any case, above a CNT width of 1.0 nm, water molecules have enough space to pass by each other, therefore single-file motion is replaced by classical (Fickian) diffusion [61,64,75,76,127,149]. Interestingly, the transition from single-file to Fickian diffusion is not spatially uniform. Several NMR studies of CNTs with sizes between 1.4 nm and 4.0 nm found *two* water components: a central water-chain diffusing in a single-file [58,63,64] and (at least) one tube-like water structure diffusing classically (corresponding to the distinct water-groups identifiable in Figure 4f–i). A second type of spatial variation of the diffusion mechanism was reported by Barati and Aluru [127] using MD simulations. They found that in large CNTs the diffusion mechanism of each water component depends on its distance from the wall. The same research group reported that in (20, 20) and (30, 30) CNTs, the mechanism of diffusion evolves from Fickian bulk-like at the center, towards ballistic for water near the wall. They also observed the evolution of the water flow from ballistic to Fickian during the first nanosecond, while water near the walls retains its ballistic character.

#### 7.3.2. Diffusion Coefficient versus the CNT’s Width

Turning to the diffusion coefficient (rate), *D*, in the pivotal MD study of Barati and Aluru [127], they studied it in a wide range of CNT diameters, from 0.95 to 6.8 nm (see Figure 7). They found that the average diffusion rate is smaller than bulk water for diameters up to 2.2 nm, as the geometrical confinement effects outweigh the H-bond reduction in regards to their effects on diffusion. This is in agreement with other studies, such as Mashl et al. [22] (MD study of CNTs with 0.31 < d < 1.81 nm), Mukherjee et al. [77,155] (MD study of 8 Å-wide CNTs) and the NMR studies of Hassan et al. [63] and Gkoura et al. [64]. Above that CNT width ( 2.2 nm), Barati and Aluru found that water diffuses faster than bulk, attaining its maximum velocity for a diameter of 2.6 nm. For wider diameters, the diffusion rate gradually drops, approaching the bulk value for diameters above 5– 6 nm. For instance, Liu et al. [61] studied with PFG NMR the diffusivity of water in wide CNTs below room temperature and they found that the diffusion rate in DWCNTs with a width of 2.3 nm was twice as big as that in MWCNTs of diameter 6.7 nm at the same temperature.

In the pursuit of forming a unified picture of how the geometry of nanosystems affects the self-diffusion of water, Chiavazzo et al. [30] argued that a simple formula with just three parameters—the bulk and ultra-confined diffusivities, $D_{B}$ and $D_{C}$, plus a unitless parameter θ characteristic of the confining geometry—adequately captures the dependency of diffusivity to the system’s size in sixty different systems, including CNTs, spherical nanoparticles, proteins, etc. In the case of CNTs, the scaling parameter θ that enters in their proposed formula is simply the ratio of the volume of water near the walls (near enough to interact with the carbon atoms), over the total volume of internal water.

#### 7.3.3. Diffusion Coefficient of Individual Water Structures

The above remarks detail the evolution of the *average* water diffusivity versus CNT width. However, as discussed in Section 5, in the intermediate range of 1.2 < d < 3.5 nm, several water components are present, each with its distinct diffusion distribution. Until recently, this interesting ensemble was studied only theoretically with MD simulations. For example, in CNTs with widths between 1.1 and 1.2 nm, Pascal et al. [110] found with MD simulations that the ice-like tubes of water (discussed in Section 5), show little in-plane diffusion, but at the same time their axial diffusivity is comparable to bulk water.

Two recent NMR studies by Hassan et al. [63] and Gkoura et al. [64] utilized 2D NMR and managed to resolve the diffusion properties of each component individually. With diffusion-relaxation (D-T${}_{2}$) and spin-lattice-spin–spin relaxation (T${}_{1}$-T${}_{2}$) spectroscopy, Hassan et al. reported that in DWCNTs with a width of 3.5 nm, the central water-chain exhibits stratified motion with faster-than-bulk diffusivity and high-fragility, whereas the surrounding water-tube has bulk-like characteristics, in agreement with previous theoretical works [74,139,140]. Their NMR study of a narrower ( 1.2 nm) SWCNT showed bulk-like diffusion, again in agreement with the theoretical predictions discussed in Section 5, which claimed that this width is too wide for single-file motion, but too narrow for the development of multiple water-components [135] (Figure 4d). The recent NMR study by Gkoura et al. [64], which studied water diffusivity across a wide range of CNT widths in a mix of SWCNTS, DWCNTS and MWCNTs, offers the first *systematic* experimental evidence that supports the aforementioned general remarks of Barati and Aluru. They found that in the diameter range of 2.5 < d < 4.5 nm multiple water components coexist in the CNTs, each acquiring a different self-diffusion coefficient, with a central water-chain exhibiting an exceptionally high velocity. The water-chain starts forming above CNT sizes of 1.1 nm and has the highest impact on the average diffusivity at about 3.0 nm, in qualitative agreement with the conclusion of Barati and Aluru about the width that maximizes the diffusivity. Above that size, the water-chain gets progressively overshadowed by more and more bulk-like water, until its contribution to the average diffusivity becomes completely negligible, especially above ∼4.5 nm.

This non-monotonic dependence of the diffusion rate on the width of the CNT was studied theoretically by Zheng et al. [160], who obtained an empirical formula that captures the competitive effects of the smooth surface and nano-confinement. According to their work, nanoconfinement is the dominant factor up to a diameter of 1.6 nm, with surface effects dominating the diffusion rate above that size.

#### 7.3.4. Discrepancies between Reported Diffusivities

Although most—if not all—studies agree with the above-discussed trend that connects the diffusivity and the CNT diameter, there are discrepancies between studies regarding the *absolute* diffusivity.

For example, the experimentally-detected diffusivity that Gkoura et al. [64] reported, was 3-fold faster than what the MD studies would predict for the same CNT diameter. Large discrepancies between studies or models in regard to diffusivity are very common in the literature. For example, even though a large number of experimental studies agree on the activation energy barrier for 1D Li diffusion in rutile TiO${}_{2}$, they disagree on the diffusion rate by even six orders of magnitude [102,153,161,162,163,164]. Another example is Li diffusion in spinel LiCoO${}_{2}$ [165,166].

To understand the source of these discrepancies in the MD studies, Alexiadis and Kassinos [75] studied the same configurations of water in CNTs with six different models (rigid or flexible CNT walls, combined with TIP3P, SPC or SPC/E water models). All these models agreed well with each other regarding the evolution of the HB-network structure, the diffusivity mode (single-file, Fickian, etc.) and the water density versus CNT diameter. Nonetheless, in regards to the *actual* diffusivity of water, all models were found to agree for narrow (d < 1 nm) CNTs, but for wider CNTs there was an offset between the diffusivities simulated with different models, which differ from one another by up to 2-fold. In this regard, the TIP3P water model yielded the fastest diffusivity (regardless of the rigidity of the CNT wall), while the combination of the SPC/E water model and a rigid-wall gave the slowest diffusion rate.

### 7.4. Water Flow in CNTs

Let us now turn to the study of water’s *flow rate* in CNTs. Flow rate is a concept closely related—but by no means synonymous—to diffusivity. On the one hand, the diffusion rate governs how fast water molecules move through a CNT of a given length; on the other hand, the flow rate states *how many* of these water molecules get transported through that CNT per unit time (i.e., number of moles/s). All other things being equal, the flow rate is proportional to the diffusivity, *for a given CNT size*. However, a wider CNT allows quite obviously more water to pass through per unit time than a narrow tube does, provided that the diffusion rate does not change dramatically between the two. Hence, the CNT diameter that maximizes the flow rate might, in principle, be very different than the 2.5– 3.0 nm size, which is found to maximize the diffusivity (see Section 7.3). This is an important distinction, as there are applications for CNTs that require maximum water flow through them and do not care about the rate of the diffusion per se (e.g., desalination), and vice versa (e.g., in nano-medicine applications).

The enhanced flow of water through CNTs is indeed one of the most interesting characteristics of these systems, at least in terms of its potential applications and its striking disagreement with macroscopic flow theory. Already back in 2003 Karla et al. [4] predicted with MD simulations that the flow rate of water through narrow CNTs (diameter of 8.1 Å) should be particularly high, comparable to that of fast biomembranes, such as aquaporin. The experimental validation of the high flow came a few years later [97], with the experiments of Majumder et al. [123] in 2005 and Holt et al. [124] in 2006. Both research groups reported flow rates orders of magnitude higher than what conventional flow rate theory would predict, the former in wide ( 7.0 nm) and the latter in narrower ( 2.0 nm) carbon nanotubes. Majumder et al.’s measured flow was four-to-five orders of magnitude larger than what the Hagen–Poiseuille equation would predict, while the flow measured by Holt et al. was estimated to be up to 8400 times larger than what that equation would estimate. Due to the scale of the discrepancies, Verweij et al. [167] argued that the classical flow models (Knudsen for gas and Poiseuille for water) should be readily abandoned in nanoscale CNTs. They attributed the failure of these models to the frictionless character of the CNT walls. Nonetheless, other groups attributed the cause of these discrepancies to different properties of the CNTs. For example, in Joseph and Aluru’s MD study of water in (16, 16) CNTs [74], the enhanced water flow is suggested to stem from the depletion region close to the CNT wall, while Walther et al. [87] showed that flow is length-dependent and its enhancement cannot be explained by the interactions of water with pristine CNT walls. Indeed, real CNTs always have impurities and defects, but their possible effects are usually neglected in theoretical studies (see Section 9).

Since models of continuous flow clearly cannot capture the discontinuous molecular flow in relatively narrow CNTs, Walter et el. [87] proposed the addition of a correction term to Hagen–Poiseuille law—put forth by Weissberg [168] in 1962—to account for membrane-end losses. This was termed as Hagen–Poiseuille–Weissberg (H-P-W) equation by Borg et al. [90], who utilized it together with MD simulations and studied the water flow in CNTs of a wide diameter range. Their results agree with several experimental studies below 4 nm, but in wider CNTs the experiments yield water flows orders of magnitude larger than what MD studies indicate, even when using the aforementioned revised theoretical tools (see Figure 8). Clearly, more work is needed in this crucial area, including systematic experimental studies of the flow versus CNT width in the whole relevant diameter range.

## 8. Exotic Ice Phases and Suppressed Water-Ice Transition Temperature in CNTs

In macroscopic settings, when the dimensions of an ice crystal are considered to be infinite, water molecules in ice obey the so-called “bulk ice rule”, where molecules are tetrahedrally coordinated, with each of them simultaneously accepting and donating two H-atoms with their neighbors [112]. Nonetheless, in nanoscale systems such as the one in question, the finite dimensions come into play and can lead to a variety of ice crystal structures. These exotic ice shapes are potentially very different compared to its bulk form. Generally speaking, nanoconfinement is known to cause phase transitions not seen in bulk, even for simple fluids consisting of small non-polar molecules [41]. The fact that water is polar, only adds to the complexity of the situation in CNTs, see Figure 9.

### 8.1. Ice Structures under Low and High Pressure

In 2001, Koga et al. [135], used MD simulations and found that water frozen in CNTs of diameter between 1.1 and 1.4 nm at temperatures below 280 K and pressure above 50 MPa (up to 500 MPa) tends to form exotic ice structures: ice-tubes made of stacked n-gonal (i.e., pentagonal, hexagonal, etc.) ice-rings, with the value of *n* = 4–6, depending on the CNT diameter. Each ice-ring has an OH group lying on its plane and another perpendicular to it and each each water molecule is tetrahedrally coordinated, as mentioned before for bulk ice [111]. The existence of the pentagonal and hexagonal ice structures was further supported by the ab initio study of Bai et al. [68] of ice tubes in vacuum and 0 K (in the sense that DFT calculates the ground state of the system, i.e., at “zero Kelvin”). In a subsequent MD simulation study, Bai et al. [169] examined in detail the ice structures inside zigzag (n, 0) CNTs at even higher pressures (up to 4.0 GPa) and they found a cornucopia of ultra-exotic ice-structures (see Figure 9). They reported six distinct ice-phases, such as double-stranded helixes in smaller-diameter (17, 0) CNTs, all the way to triple-walled helical ice with an outer wall of 18-stranded helix and the inner walls made of hextuple-stranded helixes in wider (24, 0) CNTs.

In the low-pressure regime, Ohba et al. [170] used MD simulations to show that the minimum water density to permit nano-ice structures is 0.5 g/mL. Below that critical density, the structure of water in CNTs retains its liquid characteristics at room temperature [170]. Further, Kolesnikov et al. [111] utilized ND and INS experiments, as well as MD simulations, to identify the ice-structure inside 1.4 nm CNTs under ambient pressure. They reported a central water-chain, surrounded by an ice-tube. The water-chain has a low number of hydrogen bonds—1.86 per molecule—leading to a soft dynamics and enhanced thermal motion of the central water in the transverse direction. These effects lead to a large mean-square displacement and a fluid behavior with a lower freezing temperature, compared to bulk.

Following these theoretical results, a large number of studies verified these predictions and shed light to different aspects of water dynamics and structure including the water-ice transition temperature, T${}_{ice}$, as well as the co-existence of a central water-chain and a surrounding ice-tube for certain CNT diameters. Polygonal ice-tubes inside CNTs have been reported at ambient pressure with XRD [137,138], the water-chain and ice-tube complex was reported with neutron diffraction and neutron inelastic scattering experiments [111], whereas a large number of NMR studies examined the dynamics and transition temperature of water in CNTs [49,52,53,58,60,63,64].

### 8.2. NMR Studies of the Water-Ice Transition Temperature

All in all, the above mentioned characteristics of ice in CNTs, as well as the water-ice transition temperature seem to depend on the pressure, the diameter of the tube and (possibly) the isotope of hydrogen making up the internal water (i.e., heavy or light water, see below). A particularly interesting region seems to exist for CNT diameters between 1 and 4.0 nm, where there is, at least, one ice-tube(s) relatively close to the CNT wall, plus possibly a water-chain at the center (in accordance with Figure 4). These characteristics and their dynamics have been studied extensively with nuclear magnetic resonance.

NMR is a particularly well suited experimental tool for the task at hand. Using a simple NMR lineshape analysis (see Section 3.1), the water-ice transition can be readily identified, because the transverse relaxation time T${}_{2}$ of ${}^{1}$H in ice (∼ 6 μs) is *much* shorter than that of liquid water (∼seconds). As the width of the NMR frequency peak is inversely proportional to the relaxation time (lineshape is, after all, the Fourier transformation of the NMR time-signal), this means that the water-ice phase transition abruptly makes the corresponding peak too wide to be measured (e.g., on the order of several tens of kHz). As a result, one can use the intensity of the NMR peak versus temperature to study the phase transition, as well as to distinguish between water trapped inside the CNTs and bulk water in the sample: The latter promptly freezes below 273 K and its relevant peak vanishes, whereas the internal water stays liquid *much* below the nominal bulk freezing temperature. This allows examining the dynamic and static properties of internal water without having to remove any contributions from bulk water outside the CNT channels [49]. Consequently, one can monitor the reduction of the molecular mobility upon decreasing the sample’s temperature by examining the FWHM of the NMR peaks. Fast molecular motion averages out the field felt by the spin-probes, leading to the the motional narrowing of the corresponding peak. When the molecules start to freeze below the timescale of the NMR measurement, their FWHM broadens towards their intrinsic value.

Turning first to an NMR study of narrow CNTs, Sekhaneh et al. [52] performed a high-quality magic angle spinning (MAS) ${}^{1}$H NMR lineshape experiment on samples of various CNTs with diameters ranging from 0.9 to 1.1 nm and with different chiralities. They found two spectral components, one having a chemical shift close to that of bulk water (4.6 ppm relative to TMS) that disappears below 250 K and is only present in samples overloaded with water (214 and 366 wt %) and a second one (at 1.3 ppm) that is also present in samples undersaturated with water (87 wt %) and is visible down to at least 220 K. They attributed the first component to water adsorbed at the exterior of the CNT bundles and the second component to internal water. This is in line with the discussion in Section 5, according to which at this diameter range only a single water component is expected inside the CNTs, with water diffusing in a single-file arrangement (see Figure 4b–d).

For CNTs of larger diameters (between 1.2 nm and 3.5 nm), several NMR experiments studied the water-ice temperature (T${}_{ice}$) and the characteristics of water inside the tubes. As a reminder, in this diameter-range MD simulations suggest the co-existence of a stratified central water-chain, surrounded by one (or several) water-tubes, see Section 5. For instance, the MD simulations of Kolesnikov et al. [111], accompanied by neutron diffraction (ND) and inelastic neutron scattering (INS) experiments on 1.4 nm CNTs, showed the aforementioned chain-and-tube structure. In the liquid state, we already mentioned that these water components were recently studied in great detail with 2D NMR [63,64] (see Section 7); but a large number of earlier NMR measurements had already probed the water-ice transition temperature in this diameter range using lineshape analysis.

Ghosh et al. [49] studied CNTs of 1.2 nm diameter using ${}^{1}$H NMR. Below room temperature, they identified two spectral components in the NMR lineshape, the first one appearing above 242 K and the second one vanishing below 217 K. They associated the first component with water molecules of the central water-chain, whereas the second component was noted to arise from the molecules of the water-tubes surrounding the central chain. The water-ice transition temperature of this scenario would be 242 K, because the water-tube’s spin relaxation is associated with water-ice interactions and movement of the water towards the wall, rather than with fast molecular motion and liquid character. Regarding the association of each spectral component with a particular type of internal water, note that Sekhaneh et al. [52] argued in their aforementioned MAS NMR study that one of the components seen by Ghosh et al. might stem from outside water. Nonetheless, in their case the low-field resonance peak vanished first, the opposite of what Ghosh et al. reported. Furthermore, note that the CNTs studied by Sekhaneh et al. were too narrow for an ice-tube to form, whereas the CNTs of Ghosh et al. are at the low end of that region. From the above, we conclude that there is no discrepancy between Sekhaneh et al. seeing one spectral line (excluding the external water) and Ghosh et al. finding two resonance NMR peaks.

Matsuda et al. [53] studied 1.35 nm CNTs with both ${}^{1}$H and ${}^{2}$H NMR, utilizing lineshape analysis below room temperature (between 100 and 300 K). They also reported two spectral components above 220 K, with one of them having liquid-like characteristics and exhibiting an NMR motional narrowing. The other observed component has an ice character. This study also is compatible with the chain-and-tube configuration expected in this diameter range, which is further supported by the study of Das et al. [58]. In contrast with the previously discussed experiments, Das et al. performed ${}^{1}$H PFG NMR on water in CNTs of 1.4 nm diameter and they could thus identify the diffusion mechanism of water below room temperature. According to that study, the central chain-like component freezes at 223 K and diffuses in a single-file mode, while being surrounded by tube-like ice that freezes already at 273 K. In Section 7, we saw that this single-file character of the central chain was also reported by Hassan et al. [63] and Gkoura et al. [64]. The cause of this single-file motion is a bit different than the usual confinement-based mechanism that causes *all* water inside narrow (< 1.0 nm) CNTs to diffuse in single-file. This diameter range is wide enough for water molecules to bypass each other, but the existence of the water-tube (or ice-tube below room temperature) further restricts the capillary size available for the diffusion of the central water [60,171] and therefore causes single-file diffusion for the central water-chain only.

Kyakuno et al. [60] used a combination of powder XRD, NMR, and electrical resistance measurements with different CNT sizes and found that the ice-phase behavior below and above the CNT size of 1.4 nm is quite different from each other, an indication of a crossover region. For instance, in CNTs of 2.4 nm diameter, their XRD and resistivity measurements imply that freezing water becomes unstable and a large percentage of it is ejected from the SWCNTs, with the rest forming ice nano-structures.

Turning to studies using other experimental techniques, Reiter et al. [37] examined 1.4 nm SWCNTs and 1.6 nm DWCNTs with XRD and neutron scattering. They found that the proton momentum distribution was unchanged between 4 K and 230 K, with the relevant kinetic energy up to 230 K being 30% smaller than that of bulk ice, at room temperature. Above **230K**, the kinetic energy changed significantly upon warming. These suggest that up to 230 K the protons are confined in a local Born–Oppenheimer potential. In contrast, in the DWCNT sample the proton momentum distribution varies non-monotonically between 4 K and 300 K, suggesting that the O-H bonds are stretched by 0.22 Å between 10 K and room temperature.

### 8.3. Variation of T${}_{ice}$ with CNT Width

In their 2016 review of NMR studies of water in CNTs, Hassan et al. [11] established that when all the above reported freezing temperatures of CNTs with diameters larger than 1.4 nm are plotted versus CNT size, a linear relationship is evident, with T${}_{ice}$ increasing for increasing CNT size. This trend is in agreement with studies of water in mesoporous materials, mostly MCM-41, of diameter above ∼ 1.4 nm (see Reference [60] and references therein). According to XRD measurements, the above linear relationship is reversed below that CNT width, with the freezing temperature being *inversely* proportional to the CNT diameter in the region between 1.17 nm and 1.44 nm [138,172,173]. Quite possibly, this reversal might stem from the fact that in that region there is no water-chain in the center of the CNT, only a water(ice)-tube, which generally freezes much earlier than the chain in CNTs of wider diameter (see Section 8.2).

The theoretical basis for the above phenomenological linear relationship between ice-water transition temperature suppression and CNT width in wider (> 1.4 nm) CNTs is the Gibbs–Thompson equation. This relates the suppression of the freezing temperature Δ*T* to the thermodynamic properties of the given liquid [49], based on the expression:(3)$$\Delta T=-k/R=-2\gamma MT_{0}/\left(R\rho \Delta H\right),$$
where *T*${}_{0}$ is the freezing temperature of the bulk liquid, γ its surface tension, *M* the molecule’s weight, ρ its density and Δ*H* its molar heat.

Intriguingly, the NMR studies of heavy water (i.e., using ${}^{2}$H as probes) also show a similar linear relationship, but shifted relevant to their light water counterparts higher by roughly 35 K [11]. At this point it is not clear if this difference in the freezing temperatures of heavy water is a real effect, or it stems from systematic experimental factors (e.g., samples of different quality, or not properly interpreting the electric quadrupolar effects seen by the spin-1 deuterium, but not by the spin-1/2 proton). Note, however, that the mass difference between heavy and light water molecules is ∼11% and that the phase transition temperature suppression is proportional to the molecular weight in Equation (Equation 3), suggesting that heavy water should freeze at a temperature ∼ 25 K *lower* than light water.

Turning to NMR studies that used both light and heavy water, Matsuda et al. [53] found no difference between the two isotopes, suggesting that the above discrepancy might stem from experimental factors, whereas Kyakuno et al. [60] quoted a phase transition at ∼ 220 K for ${}^{1}$H and ∼ 240 K for heavy water in 1.94 nm SWCNTs. Clearly, further NMR experiments comparing the two isotopic probes across the relevant temperature range are needed to elucidate the effect of the isotope of hydrogen on the transition temperature.

## 9. Effect of CNT Impurities, Defects and Functionalization

### 9.1. Water in CNTs under Non-Standard Conditions

All that was discussed so far detailed the characteristics of water in pristine, defect-free, uncharged, straight CNTs. Moving forward, there are quite a few variations of this fundamental system that one could study, each possibly exhibiting a distinct water character. For example, a few studies have probed the properties of water in CNTs illuminated with EM radiation (microwave and far-infrared pulse-fields [174,175,176,177,178]). In this context, Zhou et al. [179] simulated (with molecular dynamics) the water properties in (6, 6) CNTs upon illumination with pulsed EM fields. They found that the diffusivity of water decreases with increasing (axial) pulse-field frequency, as it enhances the water-wall collisions.

Another non-standard condition that has recently started to attract some scientific interst is that of CNTs with non-straight geometries. These include studying “hourglass” nanotubes [180,181] and CNT intersections and nanojunctions [182,183].

In nanojunctions, Ebrahimi et al. [183] found that the uptake dynamics are very similar to those in straight CNTs, but the wetting process in nanojunctions is much more complex than in microjunctions. Hanasaki et al. [182] studied the water flow at the junction between a wider (upstream) CNT and a narrower (downstream) tube, versus the ratio of the two diameters. They found that the existence of the junction enhances the streaming velocity, reduces the pressure and increases the temperature of water, while the ratio of the downstream-to-upstream velocities increases with the reverse ratio (upstream-to-downstream) of the two sections’ diameters.

Turning to the hourglass geometry, Graville et al. [180] found with MD simulations that a conical entrance at each side of the (usual) straight CNT enhances the water permeability, with the cone angle of 5${}^{\circ }$ being the optimum. Interestingly, their results agree quantitatively with continuum hydrodynamics, in contrast with the water flow in long, straight CNTs (see Section 7.4).

Finally, Naguib et al. [41] studied with a combination of TEM, electron energy loss spectroscopy (EELS) and energy dispersive spectrometry (EDS), as well as MD simulations, the uptake of water inside *closed* CNTs (diameters between 2 and 5 nm). In such a setting, water can sometimes penetrate the wall of the CNT through defects on the tube’ wall, a process that becomes significantly easier above the water’s critical point ( 374.14 ${}^{\circ }C$, 22.064 MPa). The maximum water occupancy they reported was 15% of the tubes (when the CNTs were fabricated with CVD) at a pressure of 80 MPa and temperature of 650 ${}^{\circ }C$. In CNTs fabricated with arc evaporation (i.e., “true” carbon nanotubes, see Section 2), the percentage of closed tubes filled with water was lower, owing to the reduced number of defects on the CNTs’ walls.

In the CNTs that did fill with water, Naguib et al. studied the water-gas interface and found that gas fills the near-wall region (due to the hydrophobicity of the wall), as well as one end of the closed tube. Turning to the liquid-gas interface, in these relatively narrow tubes it seems that the shape of the interface diverges from the clear meniscus that is observed in tubes of larger diameter (20– 100 nm), in which water behavior is consistent with its macroscopic limit.

### 9.2. Functionalized CNTs

Hitherto, the potential influence of impurities, defects and other imperfections was completely neglected. In reality, no CNT is completely free of such defects, so it is paramount to understand the extent of their impact on the various properties of the CNT and internal water. For instance, it has been shown that a single carbon defect (forming a pentagon and a heptagon instead of two hexagons) can reduce the exceptional failure stress of CNTs by up to two orders of magnitude (from 100 GPa to 1 GPa) [184].

In addition, studying functionalized CNTs (f-CNTs thereon), having different added chemical groups, can open new possibilities for further applications, either by enhancing the CNT functionalities, or by ameliorating their weaknesses and/or furthering their strengths. As an example, f-CNTs have been shown to increase the solubility dispersion of the CNT bundles, increasing thus the reactivity of individual tubes [38,185].

Nonetheless, f-CNTs have not been studied extensively so far (see Figure 10). For a long time it was imperative to first form an understanding of nominally pristine CNTs, before adding complexities on top of them, but as the previous sections made clear, at this point we do have a more-or-less informed picture of the characteristics of CNTs and their interaction with water. So the next frontier is on studying the characteristics of water inside f-CNTs, for various functional groups and impurities.

#### 9.2.1. Applications of f-CNTs

According to a number of studies (see Reference [95] and references therein), CNTs retain their electronic and mechanical properties only when a limited amount of defects or dopants are introduced. For that reason, doped and f-CNTs have been surveyed for their possible utility across a wide range of applications. Indeed, Doped CNTs have been shown to further extend their functionalities in applications such as drug delivery [187,188], biological imaging [189,190,191], gene transfer [192], protein detection [193] and several others [194].

Another important future application of f-CNTs is (suggested to be) water purification and forward osmosis [195,196], as charged functional groups on the CNTs’ tips are shown to enhance ionic selectivity and salt rejection due to electrostatic repulsion [144]. Moreover, f-CNTs are able to purify water from organic toxins and heavy metals [27].

The electrical properties of CNTs are most commonly enhanced using nitrogen and boron doping (see Ref. [194] and references therein). Nitrogen-doped graphene/CNTs composites were demonstrated to exhibit enhanced performance as supercapasitors [197], while N-CNTs are also proposed as glucose detection sensors [198]. Simultaneously doping CNTs with both N and S atoms is shown to lead to a synergistic effect of the two dopants that could facilitate their use in fuel cells [199]. CNTs doped with MgO and MgO${}_{2}$ have been studied for a potential use as anodes in lithium-ion batteries [200].

In the context of transistor electronics, pristine CNTs are usually p-type in ambient conditions, as they tend to absorb oxygen atoms on their surfaces. Nonetheless, in recent years several approaches have been utilized to realize n-type CNT-based semiconductors. These include the functionalization of the CNTs with electron-donating groups such as hydrazine, dihydronicotinamide adenine dinucleotide (NADH), benzyl viologens (BV), poly (ethylene imine) (PEI) and decamethylcobaltocene (DMC), see Reference [24] and references therein.

#### 9.2.2. Types of f-CNTs

The number of methods for doping or functionalizing CNTs with a plethora of different dopands and chemical groups is rapidly expanding [194]. In this regard, wall-defects of SWCNTs are shown to play a significant role, as they provide anchor points for further functionalization with suitable reactive groups [95]. The types of chemical processes for f-CNTs’ fabrication generally fall in three (plus one) broad categories (see Figure 11). These are: (i) defect-group functionalization (ii) covalent and (iii) non-covalent functionalization (e.g., with polymers or surfactants) attached at the outer surface of the CNT wall. Technically, one can also functionalize a CNT by inserting particles in the interior of the tube (e.g., buckyballs—C${}_{60}$ molecules) [95], but since this does not change the structure of the CNT itself and also such an action would obviously block the water flow, here we will only discuss the addition of small ions in the water as characteristic of that “extra” type of f-CNT.

Covalent functionalization with organic pendant groups is shown to both increase the solubility of the SWCNTs in solvents and also create hot-spots on the otherwise weakly reactive surface for further covalent functionalization [185,201]. Indeed, the reactivity of pristine SWCNTs is rather small, much suppressed compared to fullerene, owing to the large size of the CNTs and their smaller curvature [95]. Thus, covalent functionalization of the wall requires strongly reactive reagents, affecting the walls and forming localized defects or dopands. Common reagents used in this context are oxidizing acids [202], fluorine [203], nitrenes [204] and others [38,205,206]. This oxidation process generates copious numbers of carboxylic acid groups attached to the entrance points of the CNTs and—to a lesser extend—their walls’ exterior [95,207,208]. The earliest studies used fluorine as an oxidating agent, which was found to decisively change the CNT’s properties, turning it from conducting to insulating material above ∼ 520 K [95]. Turning to the non-covalent functionalization, it is frequently done by wrapping suitable polymers around the exterior of the CNT (creating the “snake around a log” formation of Figure 11D), or by forming non-covalent aggregates with surfactans [95]. To characterize the f-CNTs, the most commonly used techniques are NMR, AFM and absorption spectroscopy [95].

#### 9.2.3. Water in f-CNTs

Let us now turn to the impact of imperfections, functionalization and defects on the properties of nanoconfined water in CNTs. Early MD simulation studies explored their impact on the properties of nanoconfined water in CNTs indirectly, by either modulating the strength and other properties of the carbon-water van der Waals potential well or tweaking other parameters of the relevant molecular interaction. Hummer et al. [65] showed that a reduction of the corresponding potential by 0.05 kcal mol${}^{-1}$—mimicing solvent conditions—has discernible effects on water occupancy. Further, Joseph et al. [74] studied with MD simulations the effect of wall roughness on the water in the CNT by modulating the Lennard–Jones parameters. Making the wall more hydrophilic was shown to strongly reduce the water flow enhancement inside the CNTs (compared to bulk water), as it causes the free OH bonds of water near the wall to rotate (see Section 6). Majumder et al. [209] arrived at similar observations by fictitiously increasing the electrostatic water-wall interactions and by increasing the CNT wall roughness.

Other MD studies added foreign atoms (e.g., F, or O) at the interior of the CNT’s walls and probed their effect on nanoconfined water. Even though such studies are more direct than the ones discussed above, which tweaked ad hoc the interaction parameters of the water molecules, still are not realistic, as such groups are expected to attach to defects at the exterior of the wall, not in arbitrary locations of the wall’s interior. In any case, Striolo [107] found that just eight carbonyl groups can completely block the diffusion of water in (8, 8) CNTs under low hydration levels and that a handful of oxygenated sites significantly hinders the self-diffusion coefficient of nanoconfined water. Using AIMD simulations, Clark II and Paddison [210] studied the effect of fluorination of the interior of the CNT’s wall on the properties of water, in CNTs of 1.1 and 1.33 nm diameter. They found that in the fluorinated CNTs the water molecules were localized close to the wall, forming highly ordered structures that are absent in pristine CNTs. They also observed weak interactions between water and the fluorine atoms, resembling hydrogen bonds, which occured at a higher frequency in the narrower CNTs, compared to the wider tubes.

In a study of water adsorption in a more realistic f-CNT system, having a single hydroxyl group grafted at various locations of the exterior of the CNT, Wongkoblap et al. [211] found with MD simulations that this change increases the adsorption of water molecules at the exterior of the CNT (in the space between CNT tubes in a bundle), but has very little effect on water uptake inside the tube. According to this study, the only discernible effect of the hydroxyl groups on internal water is that the onset of the uptake process happens at a lower pressure (below 0.5p${}_{0}$), than in pristine CNT tubes.

On the other hand, the MD study of Gauden et al. [121] found that the addition of carbonyl groups *at the tips* of the CNT (i.e., at the entrance), significantly affects the properties of internal water: For one thing, the existence of these groups greatly increases the enthalpy of the uptake process (see Section 4). For another thing, they found that the water-carbonyl interaction acts chaotropically on the water-structures inside the tube, namely it significantly reduces the order of the internal water molecules. From the comparison of the aforementioned studies of Wongkoblap et al. and Gauden et al., it is evident that the *location* of the functional groups at the exterior of the CNTs is a very important parameter, when it comes to their effects on internal water.

Finally, charged nanotubes are proposed as electricity-driven flow pumps. Such an application requires the existence of mobile charges in the water, as the electro-osmotic flow has been shown to vanish for uncharged CNTs filled with just H${}_{2}$O [81]. Doping water in CNTs with positively charged ions (Na${}^{+}$ and K${}^{+}$) was shown by Gao et al. [212] to exhibit a maximum diffusivity at a particular cation concentration, due to the competition between the number of free OH bonds per molecule and their orientational changes. On the contrary, water diffusivity seems to decrease monotonically with ionic concentration when anions (F${}^{-}$, Cl${}^{-}$, and Br${}^{-}$) are introduced.

In comparison with the amount of information available in pristine CNTs, it is obvious that f-CNTs are a new category of systems that lack even elementary investigation so far, both in terms of their properties, relation to internal water, and possible applications.

## 10. Conclusions and Future Research Avenues

In summary, using the results of MD simulations, NMR (especially the modern method of 2D NMR, as well as MAS and PFG/SFG measurements) and several other techniques, currently we have an informed picture of water’s characteristics in pristine CNTs. Nonetheless, the clarity of the overreaching narrative underlines all the points that are still not fully understood or well studied.

### 10.1. Conclusions

Turning first to the adequately understood aspects of water’s nature inside carbon nanotubes, it is evident that despite the CNTs’ hydrophobic and restrictive nature, water readily enters the CNTs under ambient conditions, as it does for a wide range of temperature and pressure (at least above 2.1 kPa). Apparently, the free energy of water in the tubes is in all cases lower than bulk, with the process of wetting being entropy-stabilized in different CNT diameters, except in the range 1.1– 1.2 nm, in which water uptake gets stabilized due to enthalpy.

For small CNTs with diameter d < 1.0 nm, water forms a single molecular chain near the center of the CNT and diffuses in a single-file fashion due to lack of space. This has been verified with several complementary studies, theoretical and experimental. Not surprisingly, this single water component results in a lone NMR peak, which can nonetheless be readily distinguished from outside water, based on the fact that internal water freezes at a much lower temperature than its bulk counterpart, regardless of the CNT diameter.

In the next CNT diameter range, 1.0 nm < d < 1.2 nm, water molecules have more available space to bypass each other, hence single-file diffusion gets replaced by classical (i.e., Fickian) and the water-chain structure is replaced by a water-tube made of stacked water-rings. Under ambient pressure, the structure of the latter depends on the diameter of the tube, ranging from square to heptagonal configurations. The NMR spectrum yields a single resonance in this region, owing to the sole water structure in the CNTs, but diffusion measurements shows a classical diffusion mechanism, in contrast to the single-file case found in narrower tubes.

Next, for CNT diameters between 1.2 nm and 3.5 nm, there are progressively more concentric water-tubes (their number increases with CNT size), plus a central water-chain, which get reinstated after being absent in the previous diameter range. Interestingly, these distinct structures show a lot of different characteristics. For example, the central chain is found to diffuse as a single-file, while the surrounding water tube(s) diffuse classically, except for the molecules very close to the walls, which attain ballistic diffusion. With increasing CNT diameter, these effects become ever less pronounced and the characteristics of internal water approach those of the bulk liquid. Nonetheless, it seems that near the wall water still diffuses ballistically, but with increasing width, the contribution of the near-wall fraction of water becomes insignificant, literally flooded by bulk-like liquid.

Water diffusion inside CNTs larger than 2.0 nm is faster than bulk, reaching a maximum value in CNTs between 2.5 nm and 3.0 nm, before its value starts decreasing towards its bulk limit. Nanoconfined water shows a non-Arrhenius fragile nature in all CNT diameters, but its characteristics are also significantly different than the bulk liquid for widths between 3.0 nm and 4.5 nm.

Water flow in CNTs of all sizes is orders of magnitude faster than what one would theoretically predict for a liquid flowing through a nanoscale capillary. Regarding the water flow, although virtually all relevant studies found it to be between 2 and 5 orders of magnitude higher than what Hagen-Poiseuille law would predict, the reported flow rates in CNTs of similar diameters substantially disagree with each other, as can readily be seen in Figure 8.

Upon cooling, the outside water freezes first, leaving just the internal water to participate in NMR measurements. In the diameter range that supports multiple water components (i.e., above a width of 1.2 nm), the tubular water freezes much earlier than the central chain and forms ice tubes made of water-rings of various configurations (pentagonal, hexagonal, etc.), or possibly much more exotic helical forms under ultra-high pressure. The freezing temperature depends on the diameter of the CNT, increasing linearly above diameters of 1.4 nm.

### 10.2. Perspectives

In spite of the large number of studies detailing all the above characteristics of water in carbon nanotubes, there are still several issues that need to be addressed in order to validate earlier reported results or to clarify certain areas that are still not thoroughly explored.

First, it is prudent to carefully validate each model and assumption of the MD studies (LJ parameters, water model, rigidity of CNT, neglection of quantum effects, see Section 3.2). Additionally, most MD simulations treat the carbon atoms at the rims of the CNTs as non-polar, but ab initio studies show that these atoms bear partial charges, which depend on the chirality of the CNT and affect the properties of the internal water, both in terms of its structure and its diffusion [72,107]. In any case, the striking similarity of most MD conclusions with experimental data indicates that such discrepancies should not be too severe in most aspects of the simulated system. Nonetheless, there are some areas that might benefit from a more rigorous, ab initio quantum mechanical treatment. For instance, the measured diffusivity reported by Gkoura et al. [64] was a factor of 3 faster than what MD indicated it should be under identical circumstances. Given that Reiter et al. [37] found with ab initio methods that coupling of water to the longitudinal phonon modes of the CNT wall could enhance diffusivity by 300%, it is interesting to see with further studies whether these two results might be related. Note that in most MD studies discussed here, the carbon atoms of the CNT walls have fixed positions (to reduce the computational hurdle), but Alexiadis and Kassinos [75] tested the influence of CNT rigidity to the water properties and found it insignificant in all aspects *except for the diffusivity*.

Turning to the NMR measurements in particular, although most apparent discrepancies between early studies can be explained using the variations of the diameter range shown in Figure 4, the reported chemical shifts across the literature vary tremendously and irregularly, as noted by Sekhaneh et al. [52] and Hassan et al. [11]. Part of the reason behind this variation might be that in certain diameters there are (at least) two water/ice components—the chain and the tube(s)—each with its distinct dynamics. In addition, note that most of the early NMR studies did not report their chemical shifts calibrated to TMS, which makes the comparison to be difficult. A dedicated NMR study could utilize both ${}^{1}$H and ${}^{2}$H NMR in CNTs of several different diameters and establish a rigorous relationship between the chemical shifts, capillary size, water content and temperature. Furthermore, from early NMR studies it seems that deuterated water freezes 30–35 K above the water-ice transition temperature of light water under identical conditions. Piling on, unfortunately only limited NMR studies focused so far at very small CNT sizes (below ∼ 1.4 nm), in which XRD studies report the reversal of the linear relationship between the freezing-temperature and CNT diameter.

Another perhaps counter-intuitive aspect of this system is that the number of graphene walls (SW, DW, or MWCNTs) does not seem to significantly affect the properties of water in the tube, although it definitely changes many mechanical and electrical characteristics of the CNTs themselves. Indeed, in this review we tried to untangle the seemingly contradictory results between experimental studies (NMR in particular) based on the different CNT diameters surveyed in these experiments, without minding very much the difference of their samples in terms of the number of graphene walls. As an example, the systematic 2D NMR study of Gkoura et al. [64] utilized a mix of SW, DW and MWCNTs samples of various diameters, but their results were fully compatible with the MD study of Borg et al. [90], who studied the diameter-dependence of the internal water’s structure and the water flow enhancement versus CNT width *only in SWCNTs*.

Last but most important, the properties of doped and f-CNTs and their relationship with internal water should be thoroughly investigated, as hitherto there is not much work done on these systems (Figure 10). There are numerous forms of f-CNTs and for each form the treatment and level of impurities or function groups can also be varied, which significantly complicates the relevant studies. Nonetheless, from the few initial studies in this field, it seems that functional groups at the tips of the CNT are key in regulating water (and ionic) flow in and out of the tubes and that wall-defects and impurities can severely affect the mechanical and electrical characteristics of CNT. From indirect MD simulations studies, which tweaked the hydrophobicity of the CNT wall, it seems that dopands and defects on the walls of the CNT might also significantly change the diffusion and structure properties of internal water.

Given the importance of water intercalated in CNTs both as a model for pure research on complicated phenomena in various fields (physics, geology, medicine, biology) and as a basis of a tremendous amount of possible applications, further research on the effect of functionalization is required. Indeed, in the authors’ opinion, water in f-CNTs is the next frontier of research in this field.

## Figures and Tables

**Figure 1 nanomaterials-12-00174-f001:**
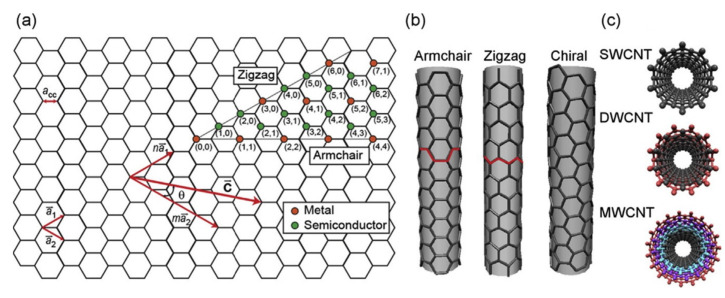
(**a**) Unrolled SWCNT showing chiral vector $\vec{C}$ and how different values of the integers n and m affect the electrical property of the SWCNT. (**b**) The direction of the chiral vector affects the appearance of the nanotube. Examples of CNTs are shown: (4, 4) armchair shape, (6, 0) zigzag shape, and (5, 3) chiral shape; and (**c**) “Ball and stick” representation of single-walled CNT (SWCNT), double-walled CNT (DWCNT), and multi-walled CNT (MWCNT) (images made using Nanotube Modeller (www.jcrystal.com)). Reprinted with permission from Reference [98]. Copyright © 2016 American Chemical Society.

**Figure 2 nanomaterials-12-00174-f002:**
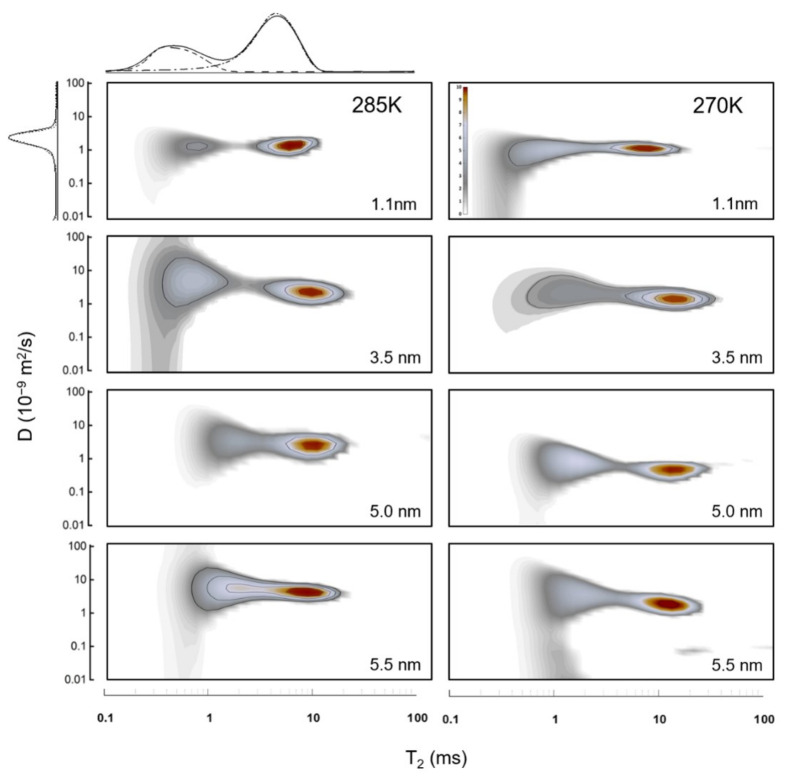
Two-dimensional (2D) ${}^{1}$H NMR D–T${}_{2eff}$ contour plots of water inside CNT of sizes 1.1, 3.5, 5.0, and 5.5 nm at selected temperatures (270 and 285 K). Note that T${}_{2eff}$ denotes the effective transverse relaxation time, which includes the effects of diffusion. Two main T${}_{2eff}$ peaks are observed, corresponding to two different water groups (interstitial and nanotube water) as seen in the T${}_{2eff}$ projection for a 1.1 nm sample at 285 K. For better visualization, all signal intensities at 270 K are multiplied by 4. Reprinted from Reference [64], with the permission of AIP Publishing.

**Figure 3 nanomaterials-12-00174-f003:**
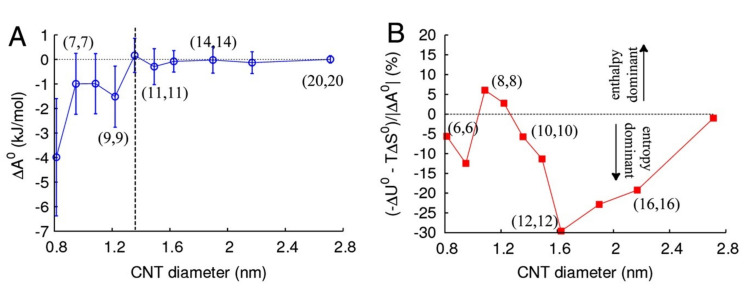
Relative thermodynamics of water confined in CNTs of infinite length. The CNTs are labeled (n, m) according to their chirality where n = m indicates armchair nanotubes. (**A**) Relative Helmholtz free energy ΔA${}^{0}$ = A${}_{confined}^{0}$− A${}_{bulk}^{0}$ as a function of CNT diameter, referenced to a bulk water box of 14,000 molecules: U${}_{bulk}^{0}$ = −34.3 ± 0.1 kJ/mol, S${}_{bulk}^{0}$ = 62.9 ± 0.2 J·mol${}^{-1}$·K${}^{-1}$, A${}_{bulk}^{0}$ = −53.3 ± 0.1 kJ/mol, where U denotes the enthalpy and S the entropy. The error bars indicate the statistical errors. The vertical dashed line indicates the point of convergence to the bulk. (**B**) Percentage of the free energy ΔA${}^{0}$ arising from the enthalpy ΔU${}^{0}$ (ΔU${}^{0}$ = ΔU${}_{confined}^{0}$-ΔU${}_{bulk}^{0}$) or entropy ΔS${}^{0}$ (ΔS${}^{0}$ = S${}_{confined}^{0}$− S${}_{bulk}^{0}$). Reprinted with permission from Ref. [110].

**Figure 4 nanomaterials-12-00174-f004:**
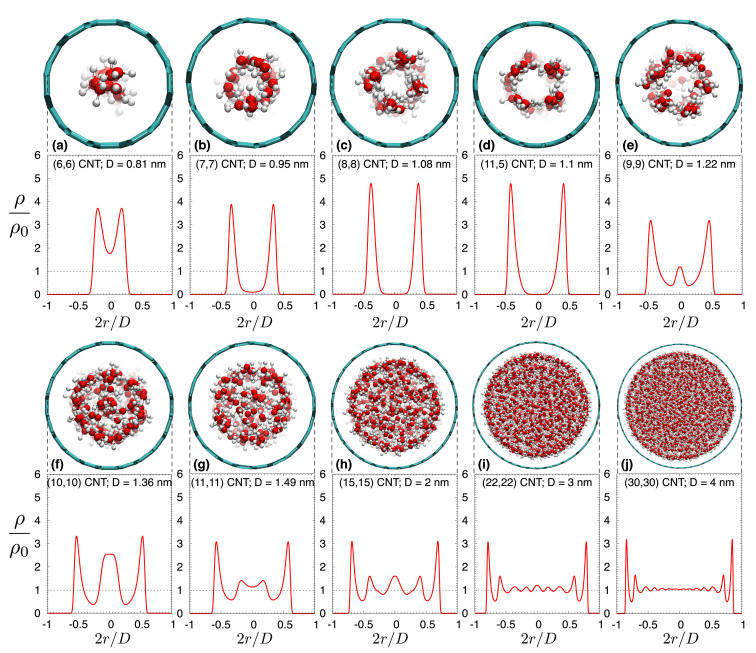
(**a**–**j**): Radial density profiles and MD cross-section snapshots of the confined water molecules at increasing CNT diameters. The axes have been normalized by bulk density $\rho_{0}$ = 1000 kg/m${}^{3}$, and carbon-to-carbon radius D/2, where *D* is the diameter of the CNT. Reprinted with permission from Ref. [90].

**Figure 5 nanomaterials-12-00174-f005:**
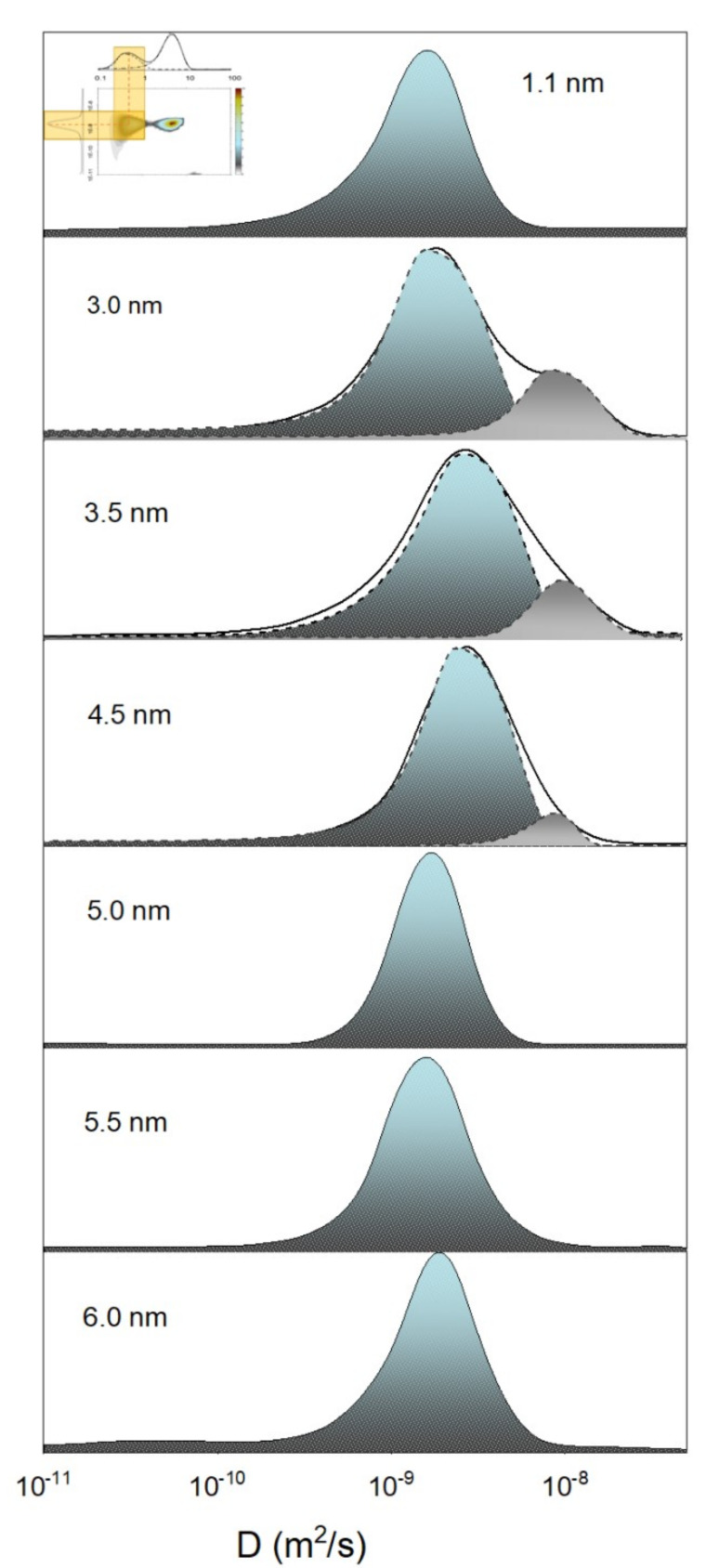
${}^{1}$H NMR diffusion projections (solid lines) from the D-T${}_{2eff}$ spectra of the internal nanotube water in different CNT sizes at 285K, where T${}_{2eff}$ denotes the effective transverse relaxation time, which includes the effects of diffusion. Diffusion projections at certain CNT sizes ( 3.0nm, 3.5nm, and 4.5nm) are resolved into two components (dashed curves), represented by the main and the shoulder peaks. Reprinted from Ref. [64], with the permission of AIP Publishing.

**Figure 6 nanomaterials-12-00174-f006:**
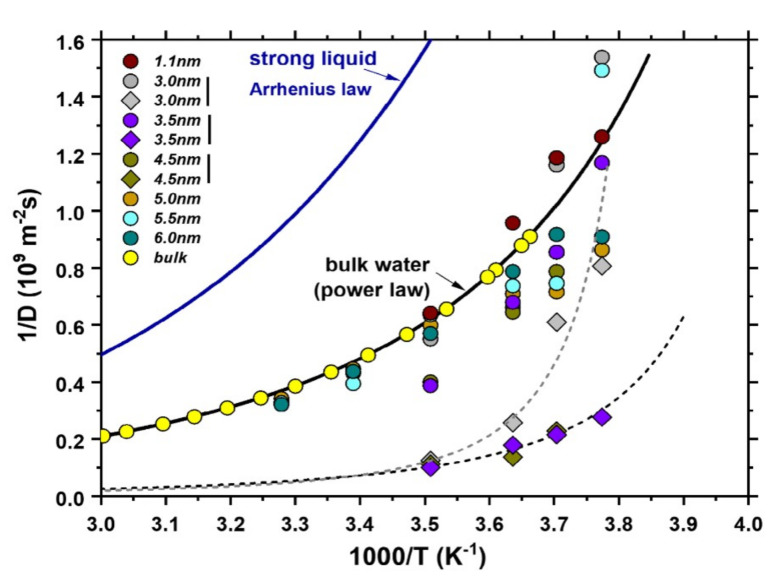
Experimental 1/D vs. 1000/T of the nanotube water in CNTs of various sizes, where *D* is the diffusivity and *T* the temperature in Kelvin. The blue lines (in both the main figure and the inset) are theoretical 1/D vs. 1000/T curves of an ideal “strong” liquid obeying the Arrhenius law. The yellow circles and the black line are the experimental values of bulk water and the relevant power-law fit. In CNT sizes of 3.0, 3.5, and 4.5nm, two water groups are resolved with different dynamics (slow and fast). The gray and the black dashed-lines are the power line fits of the data of the fast nanotube water group. The blue arrows are the relevant liquidus temperatures T${}_{l}$. The inset is magnification of the 1/D vs. 1000/T curves of the fast water component for CNT sizes 3.0, 3.5, and 4.5nm. Reprinted from Ref. [64], with the permission of AIP Publishing.

**Figure 7 nanomaterials-12-00174-f007:**
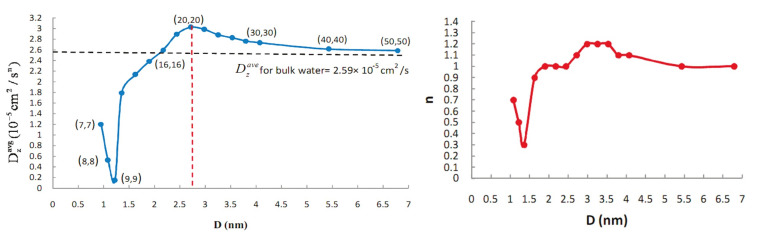
(**Left**) Variation of the average axial diffusion coefficient, D${}_{z}^{ave}$, of water as a function of the diameter of the CNT. (**Right**) Variation of n, the exponent which defines the diffusion mechanism (see Equation (Equation 2)), as a function of the CNT diameter. Reprinted with permission from Reference [127]. Copyright © 2011 American Chemical Society.

**Figure 8 nanomaterials-12-00174-f008:**
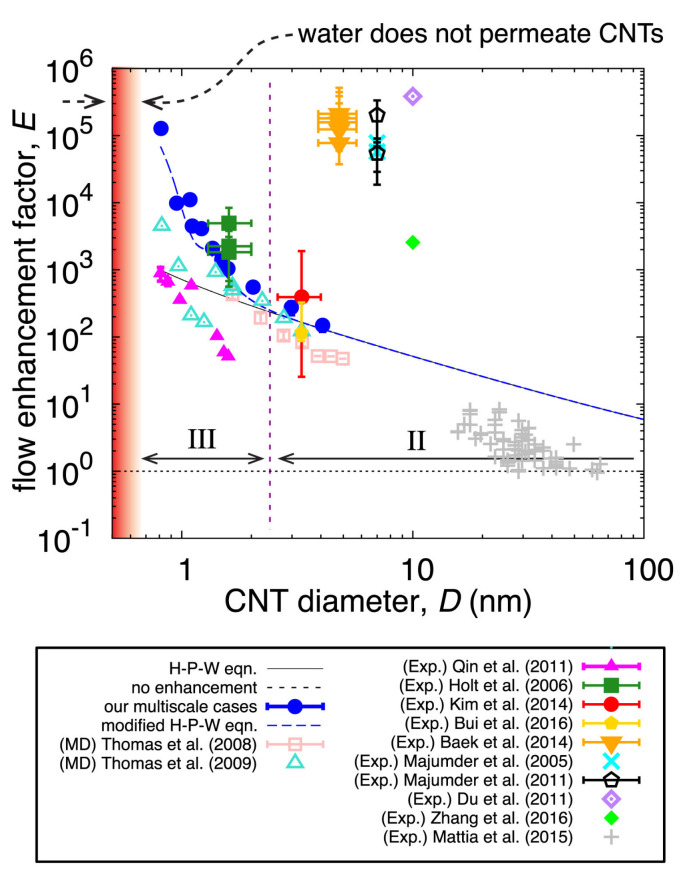
Dependence of the flow enhancement *E* for long CNTs with diameter *D*. Comparisons between Borg et al.’s multiscale MD results, their calibrated H-P-W equation, the unmodified H-P-W equation, other full MD simulations and flow experiments (see Reference [90] and references therein). Borg et al. divided the figure into three regimes: *Regime I* (not shown) where no-slip flow equations can be used (D ≥ 1μm), *Regime II*, where fixed slip-flow can be used, and *Regime III* (D ≤ 2nm) where diameter-dependent slippage must be accounted for. Reprinted with permission from Ref. [90].

**Figure 9 nanomaterials-12-00174-f009:**
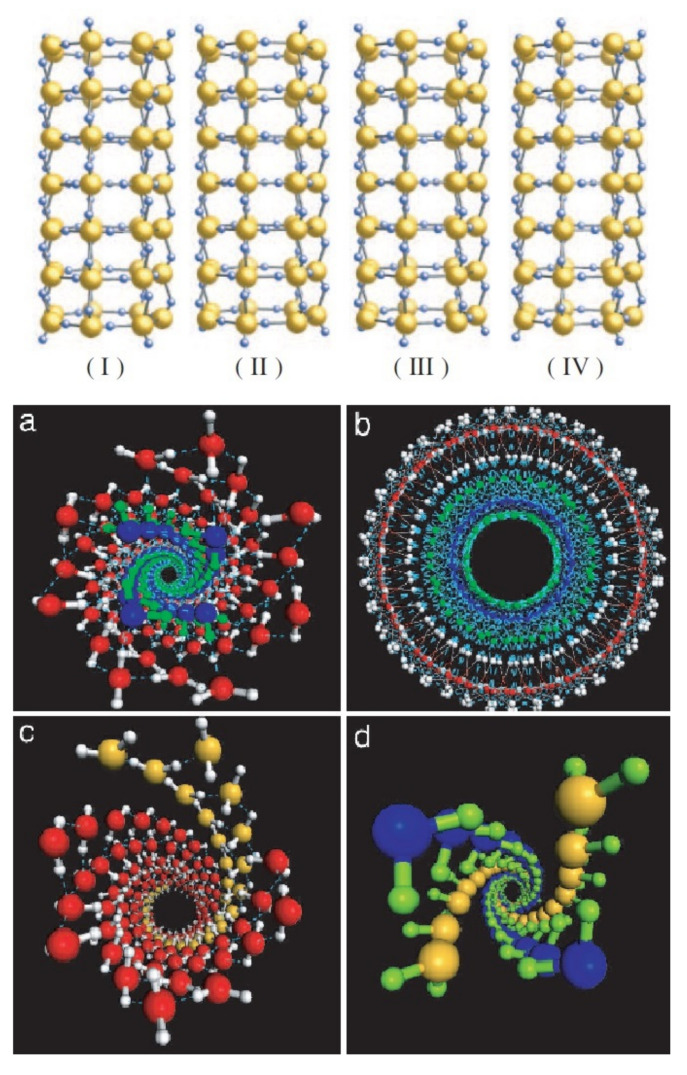
(**Up**) Schematic illustration of the heptagonal ice-NTs with energetically equivalent proton arrangements (I–IV). Larger spheres represent oxygen atoms and smaller spheres represent hydrogen atoms. Reprinted with permission from Ref. [53]. Copyright © 2006 by the American Physical Society. (**Down**) Snapshots of quenched molecular coordinates of the helical nano-ice formed in (17, 0) CNT at 4GPa axial pressure. (**a**) Top view of the double-walled nano-ice helix in the axial direction. Water molecules in the outer wall are in red–white, whereas those in the inner wall are in blue–green (the blue dashed lines denote hydrogen bonds). (**b**) Projected top view in the axial direction. Due to the helicity, the projected top view shows ring-like outer and inner wall structures. (**c**) The outer wall: an octuple-stranded helix consisting of four double helixes (one of the four is highlighted by gold). (**d**) The inner wall: a quadruple-stranded helix where two strands (gold) are proton donors and two (blue) are proton acceptors to molecules of the outer wall. Reprinted with permission from Ref. [169]. Copyright © 2006 National Academy of Sciences.

**Figure 10 nanomaterials-12-00174-f010:**
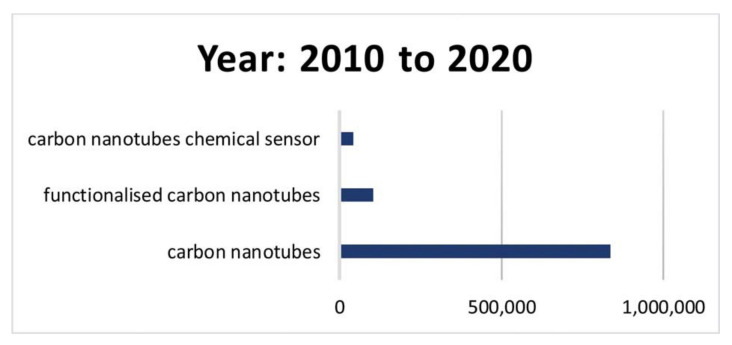
Number of publications with the specific keywords from the year 2010 to 2020. Reprinted with permission from Reference [186].

**Figure 11 nanomaterials-12-00174-f011:**
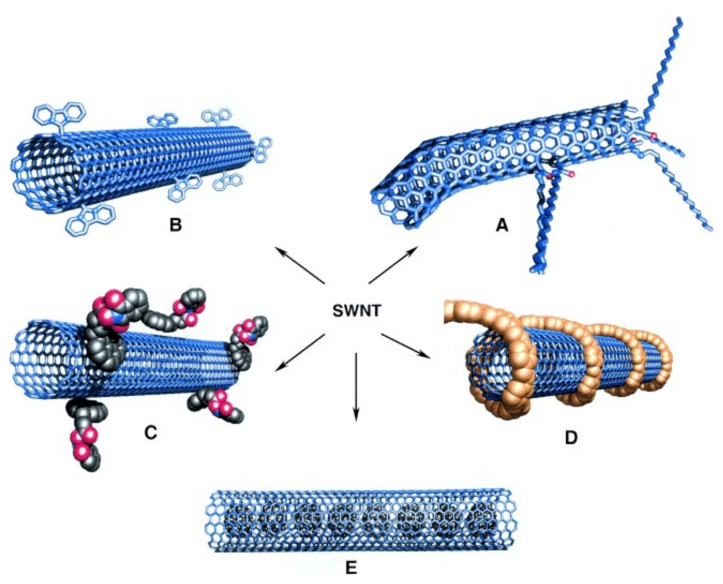
Functionalization possibilities for SWNTs: (**A**) defect-group functionalization, (**B**) covalent sidewall functionalization, (**C**) noncovalent exohedral functionalization with surfactants, (**D**) noncovalent exohedral functionalization with polymers, and (**E**) endohedral functionalization with, for example, C${}_{60}$. For methods B-E, the tubes are drawn in idealized fashion, but defects are found in real situations. Reprinted with permission from Ref. [95].

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
