# Peer review of "Current Understanding of Water Properties inside Carbon Nanotubes"

_nanomaterials, 2022, doi:10.3390/nano12010174_

Round 1

Reviewer 1 Report

The paper presents research on the current understanding of water properties inside carbon nanotubes. The presentation of methods and scientific results in the current form is satisfactory for publication in the Nanomaterials journal. The minor and significant drawbacks to be addressed can be specified as follows:
1.    The authors omitted aspects related to the existence of free space in nanotubes in contact with liquid water/ice – see papers published by Naguib, Gogotsi and co-workers.
2.    In my opinion, the following essential works were omitted:     https://doi.org/10.1039/C0CP02028A    https://doi.org/10.1063/1.1924697    https://doi.org/10.4186/ej.2013.17.4.93
3.    “3. Methods for studying the properties of water in CNTs. “The authors omitted the following techniques: differential scanning calorimetry, dielectric relaxation spectroscopy, neutron diffraction, and adsorption (calorimetry). See - https://doi.org/10.1039/B808246D
4.    Lines 381 and 382. “density functional theory (DFT) [67,106]”. DFT in the section and this paragraph on MD?
5.    “9. Effect of CNT impurities, defects and functionalization”. I think this section is a bit of “a foreign body”. There is not enough water in “9” outside 9.2.3.
6.    Conclusions are too long
7.    “10. Conclusions and future research avenues.” The section should be divided into “Conclusions” and “Perspectives.”
8.    It is a pity that the authors of the description of the experiment focused on NMR measurements.

Author Response

The authors would like to thank the reviewers for taking time to read our submitted article and put forward their constructive comments and recommendations that made the revised version of the manuscript better.

We have taken into account the reviewers’ comments in working with the revised version of the manuscript. Changes that were made in the manuscript are now shown in color in the revised version; added texts are shown in green color and deleted texts are shown in red color.

Please see the attached pdf revised version of the detail.

Below are our responses to the reviewers’ comments:

Reviewer 1:

Reviewer comment: The authors omitted aspects related to the existence of free space in nanotubes in contact with liquid water/ice – see papers published by Naguib, Gogotsi and co-workers.

Authors’ response: A discussion of the above work is added in Section 9.1 of the revised manuscript.

Reviewer comment: In my opinion, the following essential works were omitted:     https://doi.org/10.1039/C0CP02028A    https://doi.org/10.1063/1.1924697    https://doi.org/10.4186/ej.2013.17.4.93

Authors’ response: A summary of the conclusion of the aforementioned studies are now included, and cited, in the revised version of the manuscript.  

Reviewer comment: “3. Methods for studying the properties of water in CNTs. “The authors omitted the following techniques: differential scanning calorimetry, dielectric relaxation spectroscopy, neutron diffraction, and adsorption (calorimetry). See - https://doi.org/10.1039/B808246D

Authors’ response: We have included these three important experimental techniques in the introduction of the revised version of the manuscript.

Reviewer comment: Lines 381 and 382. “density functional theory (DFT) [67,106]”. DFT in the section and this paragraph on MD?

Authors’ response: In the section discussing Molecular Dynamics, DFT is mentioned mostly for contrast/comparison between these two important theoretical techniques: MD can robustly simulate dynamic aspects of large systems, while DFT only yields the ground state and is very computationally intensive, but on the other hand takes Quantum Mechanics into account. Note that DFT here is mentioned as “a powerful computational technique, not as a branch of MD.

Reviewer comment: “9. Effect of CNT impurities, defects and functionalization”. I think this section is a bit of “a foreign body”. There is not enough water in “9” outside 9.2.3.

Authors’ response: We agree with the reviewer. The authors removed some parts of Section 9.2 that were only tangentially related to water in CNTs, nevertheless Section 9.1 and 9.2.3 discuss the properties of water in these systems, therefore were left intact in the revised version of the manuscript.

Reviewer comment: Conclusions are too long

Authors’ response: The conclusion section in the revised version significantly shortened as recommended by the reviewer, without omitting the main points that are addressed in the review.

Reviewer comment: “10. Conclusions and future research avenues.” The section should be divided into “Conclusions” and “Perspectives.”

Authors’ response: the conclusion section in the revised version of the manuscript is split in two subsections, as suggested by the reviewer.  

Reviewer comment: It is a pity that the authors of the description of the experiment focused on NMR measurements.

Authors’ response: The authors feel that the ability of NMR to provide us with a very detailed local picture couples best with MD simulations in the pursuit to form a spherical understanding of the system in question.

Best regards,

Reviewer 2 Report

This review summarizes water in CNT studied with experiments and MD computations.

This topic may be important as both nanomaterials and behavior of basic substance, water.

Many literatures are including reasonable story without serious wrong description.

Therefore, it should be accepted.

However, to help readers' understanding, please insert some (conceptual) figures or schemes that stand for topics of each section.

At present, almost contents were sentences.

That's all. 

Author Response

The authors would like to thank the reviewers for taking time to read our submitted article and put forward their constructive comments and recommendations that made the revised version of the manuscript better.

We have taken into account the reviewers’ comments in working with the revised version of the manuscript. Changes that were made in the manuscript are now shown in color in the revised version; added texts are shown in green color and deleted texts are shown in red color.

Please see the attached pdf revised version of the detail.

Below are our responses to the reviewers’ comments:

Reviewer 2:

Reviewer comment: This topic may be important as both nanomaterials and behavior of basic substance, water.

Many literatures are including reasonable story without serious wrong description.

Therefore, it should be accepted.

Authors’ response: We thank the reviewer for this comment.

Reviewer comment: However, to help readers' understanding, please insert some (conceptual) figures or schemes that stand for topics of each section.

At present, almost contents were sentences.

Authors’ response: The manuscript has nine sections with eleven figures. Some of them have more than one sub-figure inside. Figure 3 and figure 7, each has figure a and figure b. We think adding an additional figure to every section will add no further information to the reader. We have kept the number of figurers to eleven, in the revised manuscript.

Attached is the revised version of the manuscript.

Best regards,
